# Phosphorylation and O-GlcNAcylation at the same α-synuclein site generate distinct fibril structures

Jinjian Hu[1,8], Wencheng Xia[2,8], Shuyi Zeng[3,4,8], Yeh-Jun Lim [1,8], Youqi Tao[3,4], Yunpeng Sun [2], Lang Zhao[1], Haosen Wang[1], Weidong Le [5,6], Dan Li [3,4], Shengnan Zhang[2], Cong Liu [2,7] ✉ & Yan-Mei Li [1] ✉

α-Synuclein forms amyloid fibrils that are critical in the progression of Parkinson's disease and serves as the pathological hallmark of this condition. Different posttranslational modifications have been identified at multiple sites of α-synuclein, influencing its conformation, aggregation and function. Here, we investigate how disease-related phosphorylation and O-GlcNAcylation at the same α-synuclein site (S87) affect fibril structure and neuropathology. Using semi-synthesis, we obtained homogenous α-synuclein monomer with site-specific phosphorylation (pS87) and O-GlcNAcylation (gS87) at S87, respectively. Cryo-EM revealed that pS87 and gS87 α-synuclein form two distinct fibril structures. The GlcNAc situated at S87 establishes interactions with K80 and E61, inducing a unique iron-like fold with the GlcNAc molecule on the iron handle. Phosphorylation at the same site prevents a lengthy C-terminal region including residues 73 to 140 from incorporating into the fibril core due to electrostatic repulsion. Instead, the N-terminal half of the fibril (1–72) takes on an arch-like fibril structure. We further show that both pS87 and gS87 α-synuclein fibrils display reduced neurotoxicity and propagation activity compared with unmodified α-synuclein fibrils. Our findings demonstrate that different posttranslational modifications at the same site can produce distinct fibril structures, which emphasizes link between posttranslational modifications and amyloid fibril formation and pathology.

Pathological aggregation of amyloid proteins such as α-synuclein (α-syn) and Tau is strongly linked to neurodegenerative diseases (NDs)[1–3]. Numerous posttranslational modifications (PTMs), such as phosphorylation, O-glycosylation, ubiquitination, and acetylation, have been identified on amyloid proteins, playing diverse roles in regulating protein conformation, aggregation kinetics, fibril structures and pathology[4–10]. α-Syn is the key player in Parkinson's disease (PD), with its amyloid fibril being the main component of the Lewy bodies (LBs)

[1]Key Laboratory of Bioorganic Phosphorus Chemistry and Chemical Biology (Ministry of Education), Department of Chemistry, Tsinghua University, Beijing 100084, China. [2]Interdisciplinary Research Center on Biology and Chemistry, Shanghai Institute of Organic Chemistry, Chinese Academy of Sciences, Shanghai 201210, China. [3]Bio-X Institutes, Key Laboratory for the Genetics of Developmental and Neuropsychiatric Disorders (Ministry of Education), Shanghai Jiao Tong University, Shanghai 200030, China. [4]Zhangjiang Institute for Advanced Study, Shanghai Jiao Tong University, Shanghai 200240, China. [5]Shanghai University of Medicine & Health Sciences Affiliated Zhoupu Hospital, Shanghai 201318, China. [6]Center for Clinical and Translational Medicine, Shanghai University of Medicine and Health Sciences, Shanghai 201318, China. [7]State Key Laboratory of Chemical Biology, Shanghai Institute of Organic Chemistry, Chinese Academy of Sciences, Shanghai 200032, China. [8]These authors contributed equally: Jinjian Hu, Wencheng Xia, Shuyi Zeng, Yeh-Jun Lim. ✉e-mail: liulab@sioc.ac.cn; liym@mail.tsinghua.edu.cn

and Lewy neurites (LNs), hallmarks of PD[11–13]. Elevated α-syn phosphorylation level, including pY39, pS87, and pS129, has been observed in the brains of PD and multiple system atrophy (MSA) patients[14–18]. While pY39 and pS129 are dominant PTMs in LBs and LNs, pS87 is not enriched[14,16,19]. Moreover, unlike pY39 and pS129 which enhances α-syn transmission and pathology in cells[7,20], pS87 has been found to prevent α-syn aggregation and neurotoxicity[15,21]. This suggests that phosphorylation at different α-syn sites may have distinct or even opposing roles in regulating α-syn aggregation and pathology.

In addition to phosphorylation, amyloid proteins such as Tau and α-syn are found to be O-GlcNAcylated[22–24], which may prevent their amyloid aggregation[8,25]. Interestingly, there is a noticeable reduction in the overall O-GlcNAc levels in brains affected by Alzheimer's Disease[26,27]. Neurodegeneration in mice is also observed when the overall O-GlcNAc level is diminished through the suppression of O-GlcNAc transferase[28]. This suggests that O-GlcNAcylation could have a protective function in NDs. It's worth mentioning that α-syn was observed to be O-GlcNAcylated at several points including T72, T75, T81, and S87[23,29,30], and this consistently demonstrated protection against α-syn aggregation[8,25,31–33]. Notably, T75 and S87 are sites of both phosphorylation and O-GlcNAcylation[15,29,30]. However, the specific effects of different PTMs at the same α-synuclein sites in influencing amyloid aggregation and neuropathology are not yet fully understood.

In this study, we focus on α-syn S87, which can be modified by both phosphorylation and O-GlcNAcylation. Employing a semi-synthesis approach, we prepared pS87 and gS87 α-syn monomer with high purity. Remarkably, we revealed by using cryo-EM that pS87 α-syn and gS87 α-syn form amyloid fibrils with distinct structures. In gS87 α-syn fibril structure, GlcNAc at S87 establishes new interactions with neighboring residues in the C-terminal of the non-amyloid-β component (NAC) of α-syn, inducing a iron-like structure. While, in pS87 α-syn fibril structure, phosphate group excludes the C-terminal of α-syn NAC from forming fibril core. Moreover, both pS87 α-syn and gS87 α-syn fibrils exhibit reduced neuropathology compared to the unmodified Wild-type (WT) α-syn fibril formed under the same condition. Our findings provide structural basis of how different PTMs lead to the formation of distinct fibril structures with attenuated neurotoxicity.

## Results

### Semi-synthesis of pS87 and gS87 α-syn

We first sought to obtain site-specific pS87 and gS87 α-syn with high purity using native chemical ligation strategy using peptide hydrazides developed by Liu and co-workers[34,35]. Since S87 is in the middle of α-syn, we performed N-to-C sequential native chemical ligation using three segments referring to the work of Pratt and co-workers[8,32,36], including segment X (α-syn 1-84 thioester), segment Y (gS87/pS87 α-syn A85C-90NHNH₂), segment Z (α-syn A91C-140) (Fig. 1a).

To obtain segment X, α-syn 1-84 fused with intein was expressed, then followed by thiolysis. Segment α-syn Y-gS87 and α-syn Y-pS87 were manually synthesized by Fmoc-based solid phase peptide synthesis. The glycosylated Fmoc-amino acid, Fmoc-L-Ser(GlcNAc(Ac)₃-β-D)-OH was synthesized referring to the work of Pratt[37]. The recombinant segment Z was expressed in *E. coli*. Methylhydroxylamine was then used to reverse N-terminal cysteine modification during expression.[38] Segment Y and Z were further purified and lyophilized for next reaction.

After obtaining these segments, we firstly ligated X and Y using native chemical ligation[39], followed by purifying. The resulting segment XY-gS87/pS87 (gS87/pS87 α-syn 1-90NHNH₂, A85C) was lyophilized and ligated with segment Z through protein hydrazide. After purification and lyophilization, XYZ with gS87/pS87 modification (gS87/pS87 α-syn, A85C A91C) was obtained. To remove sulfhydryl groups on the cysteine residues at the ligation sites, radical-catalyzed desulfurization[40] was performed to obtain α-syn protein only with

modification at S87. All segments during synthesis and the final modified protein products were characterized with analytical RP-HPLC and ESI-MS (Supplementary Fig. 2-4).

### Characterization of the gS87 and pS87 α-syn fibrils

We next investigated the influence of glycosylation and phosphorylation at S87 on α-syn fibrillation through thioflavin T (ThT) kinetic assay and negative-staining (NS) transmission electron microscopy (TEM). As shown in Fig. 1b, both gS87 and pS87 α-syn exhibited significantly reduced capability for fibrillation compared to unmodified WT α-syn. In the presence of pre-formed fibrils (PFFs) formed by unmodified α-syn protein, unmodified monomer rapidly formed fibril without nucleation (Fig. 1b). In sharp contrast, gS87 α-syn started to form fibril after 20 h incubation. The ThT signal of pS87 sample slowly picked up after 40 h with much less fibril formed as revealed by NS-TEM (Fig. 1b, Supplementary Fig. 9a). More interestingly, atomic force microscopy (AFM) revealed that both the gS87 α-syn and pS87 α-syn fibrils feature a right-handed helical twist, which is distinct from the left-handed unmodified WT α-syn fibril (Fig. 1c). Statistical analysis of AFM data revealed that the average half-pitch lengths of gS87 α-syn and pS87 α-syn fibrils are approximately 156 nm and 157 nm (Supplementary Fig. 9c, d). Additionally, we assessed the stability of these fibrils under proteinase K (PK) digestion. The results demonstrated that both gS87 and pS87 variants were digested more rapidly compared to the unmodified WT PFFs. Notably, pS87 fibrils exhibited the lowest stability (Supplementary Fig. 9b). Together, our results demonstrate that both glycosylation and phosphorylation at S87 severely impair α-syn fibrillation, and induce α-syn to form distinct morphological fibrils from unmodified fibril.

### Cryo-EM structure determination of gS87 and pS87 α-syn fibrils

To investigate whether glycosylation and phosphorylation at S87 alters the structures of α-syn amyloid fibrils, we set out to determine the atomic structure of gS87 and pS87 α-syn fibril by using cryo-EM. The cryo-EM data were collected on a Titan Krios G4 cryo-transmission electron microscope (2134 micrographs for gS87 and 2423 micrographs for pS87). 21,328 fibrils from gS87 dataset and 27,806 fibrils from pS87 dataset were picked for the following two-dimensional (2D) classification and 3D reconstruction (Supplementary Table 1).

For gS87 sample, 2D classification results reveal there are two fibril polymorphs in the gS87 α-syn fibril, including the double filament polymorph (-56%) and the single filament polymorph (-44%) (Supplementary Fig. 5a). After performing the 3D reconstruction, we managed to obtain the density map of the double filament with the overall resolution of density map of 3.1 Å (Supplementary Fig. 5c). The double filament of gS87 fibril features a half pitch (a whole pitch represents the length of a 360° helical turn of the entire fibril) of -156 nm, a helical rise of 2.41 Å, and a helical twist of −179.72° (Fig. 2a). gS87 α-syn adopts a similar architecture in both the double filament and single filament polymorphs (Supplementary Fig. 5a). For pS87 sample, 2D classification identified two different fibril morphologies, including the straight filament polymorph (-74%) and the twisted filament polymorph (-26%) (Supplementary Fig. 5b). Although the structural elucidation of straight fibrils remains challenging due to the current limitations of cryo-EM helical reconstruction techniques[41–43], we observed a distinct difference in the 2D class averages between the pS87 straight filament polymorph and the WT polymorph 1a (Supplementary Fig. 7a, b). Besides, we managed to obtain the high-quality density map of the twisted filament with an overall resolution of 2.6 Å (Supplementary Fig. 5d). The pS87 twisted filament features a half pitch of -154 nm, a helical rise of 2.41 Å, and a helical twist of −179.72° (Fig. 2b).

### Structural analysis of gS87 and pS87 α-syn fibrils

Utilizing high-quality cryo-EM density maps, we constructed two structural models for the double filament polymorph of gS87

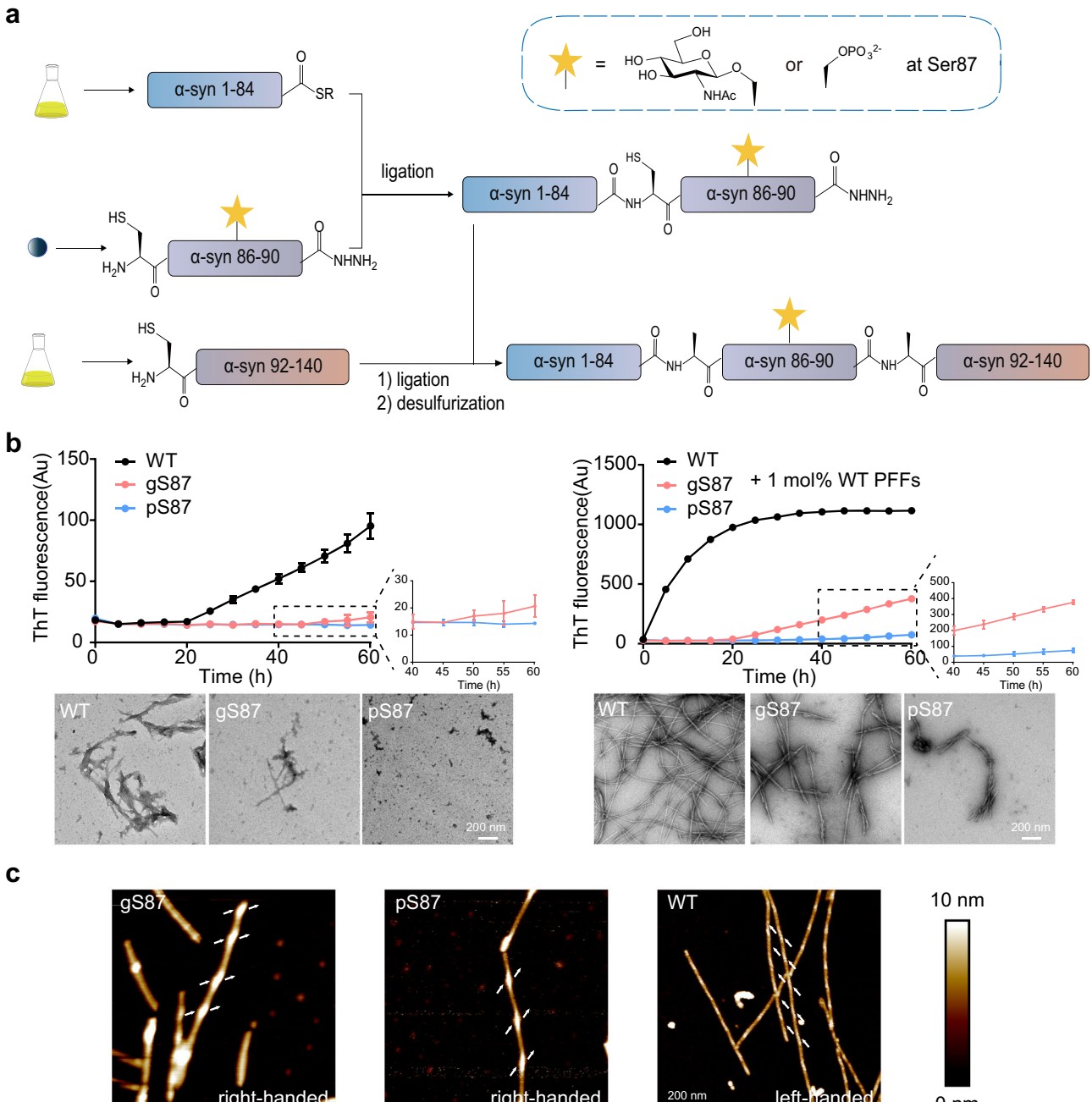

**Fig. 1 | Synthesis workflow and fibril characterization of gS87, pS87 and unmodified WT α-syn. a** Workflow of the semi-synthesis of α-syn with different modifications at S87. **b** Left: ThT kinetic assay (top) and NS-TEM images (bottom) of unmodified WT, gS87, and pS87 α-syn fibrils. Right: ThT kinetic assay (top) and NS-TEM images (bottom) of unmodified WT, gS87 and pS87 α-syn fibrils in the presence of 1 mol% PFF formed by the unmodified WT α-syn monomer. Zoom-in views of gS87 and pS87 ThT kinetic assay were shown. The fibrils were characterized by

NS-TEM at the endpoint (60 h) of the ThT kinetic assay. Data correspond to mean ± s.d., $n = 3$ (replicates). Scale bar: 200 nm. **c** AFM images of gS87 α-syn, pS87 α-syn, and unmodified WT α-syn fibril. The arrows at both sides of the fibril indicate the starting points of the fibril protrusions to clarify the handedness. Fibrils formed by gS87, pS87 and WT in >3 independent experiments provide reproducible images and their data were analyzed in Nanoscope software. Scale bar: 200 nm. Source data are provided as a Source Data file.

(hereafter referred to as gS87 α-syn fibril) and the twisted filament polymorph of pS87 (hereafter referred to as pS87 α-syn fibril). Remarkably, α-syn exhibits two entirely distinct structural conformations in these two fibril structures (Fig. 2c, d). In the gS87 α-syn fibril, α-syn adopts a conserved iron-like fold in the core structure with the modified GlcNAc molecule on the iron handle (Fig. 2c). The superior electron density of GlcNAc suggests its high stability within the core structure (Fig. 2e). The GlcNAc molecules stack on top of each other and align along the fibril axis with an intermolecular distance of 4.82 Å. The fibril core comprises residues E35 to A89, which fold into 4 β-

strands (β1-β4) (Fig. 2f). An unassigned island is observed on the outer surface of the fibril core, adjacent to β2 (residues 48-57, Supplementary Fig. 6a). This island is located approximately 41 Å from the C-terminus of residue A89, which is hypothesized to represent a segment of the C-terminal of α-syn (Supplementary Fig. 6b). The two identical protofilaments intertwine to form the gS87 fibril (Fig. 2c). Intriguingly, unlike the tight and complementary protofilament interface seen in earlier fibril structures[7,44-47], neighboring α-syn chains from the two protofilament sets remain separate, without any direct interaction. The distance between the nearest side chains of the two α-

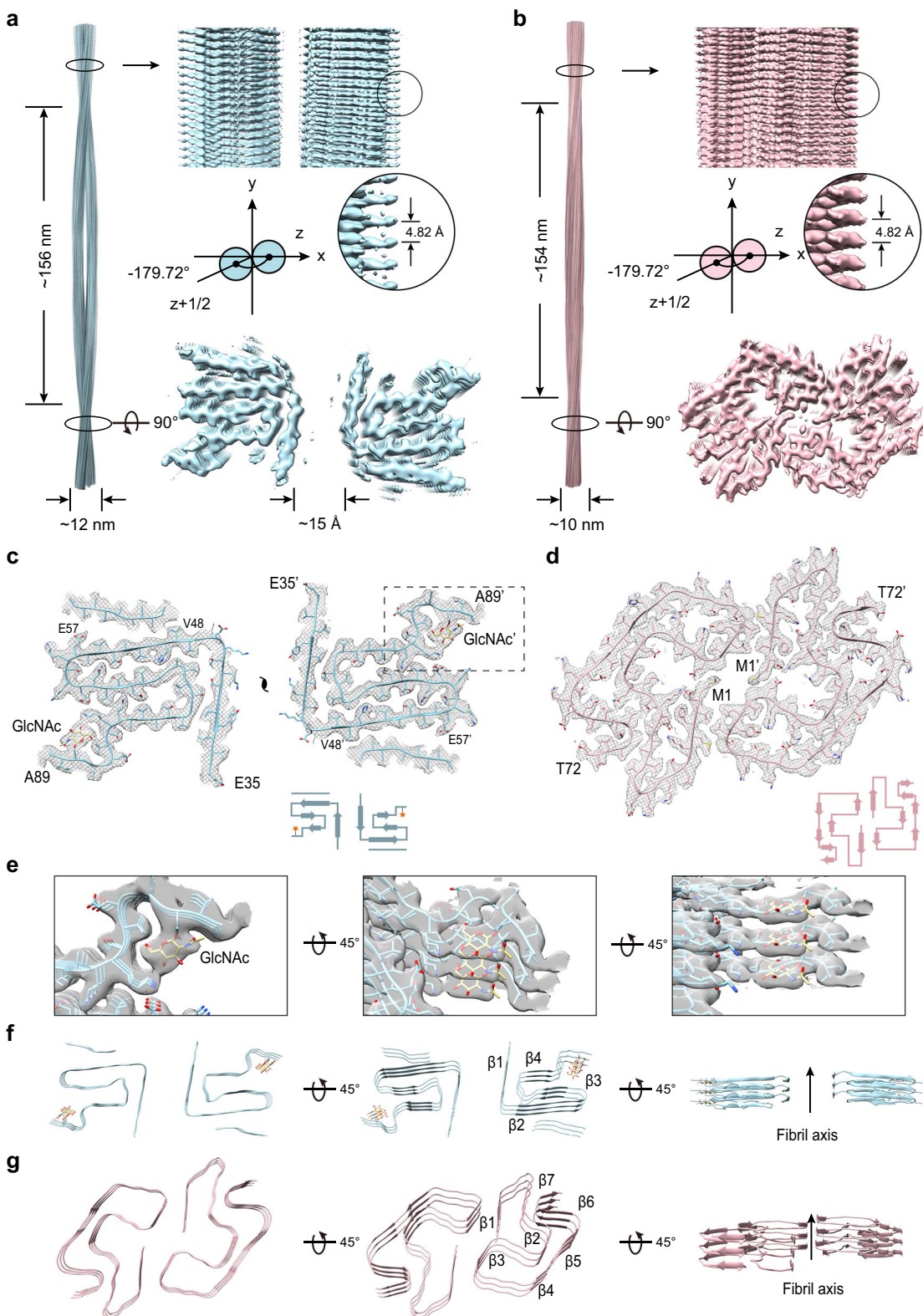

**Fig. 2 | Cryo-EM structures of gS87 and pS87 fibrils.** The density maps of the gS87 (**a**) and pS87 (**b**) fibril are colored in light-blue and pink, respectively. Fibril parameters including half-pitch, fibril width, twist angle, and helical rise are marked. Cross-section view for the density maps with a built-in structure model of gS87 (**c**) and pS87 (**d**) α-syn. Topology diagrams are shown at the bottom right. **e** Zoom-in views of the GlcNAc molecules in the electron density from (**c**) are shown. Views of three layers of gS87 (**f**) and pS87 (**g**) α-syn fibrils are shown in the cartoon. The β-strands of the fibril structures are numbered and labeled accordingly with the fibril axis indicated.

syn molecules exceeds ~15 Å (Fig. 2a), suggesting a relatively weak interaction between the two protofilaments. Two additional, albeit weak, densities were identified at the protofilamental interface (Supplementary Fig. 6c), which are hypothesized to represent solvent molecules that potentially bridge the interface together.

In the pS87 α-syn fibril, α-syn adopts an arch-like fold in the fibril core, composed of residues M1 to T72 folding into 7 β-strands (β1-β7) (Fig. 2g). Unlike the gS87 and unmodified WT α-syn fibrils formed under identical condition[44], the entire N-terminal region participates in fibril core formation in the pS87 fibril (Fig. 2d). While, the phosphorylated S87 is excluded from the fibril core, and remains flexible and invisible from cryo-EM. Furthermore, the two protofilaments of pS87 α-syn are zipped together through extensive intermolecular interactions involving the side chains of residues V3, M5, L38, and V40 (Fig. 2d). In summary, although phosphorylation and O-glycosylation modify the same residue, they lead to substantially distinct conformation of the α-syn subunit and protofilament arrangement in the fibril structures.

## Comparison of α-syn fibril structures with two different modifications

In the gS87 α-syn fibrils, the GlcNAc installed at S87 participates in fibril core formation and is located at the C-terminus of the core structure (Fig. 2c). GlcNAc forms direct interactions with the K80, T81, V82, I88 and A89 (Fig. 3a). Concurrently, K80 establishes a salt bridge with E61 (Fig. 3a), stabilizing the U-shaped structure formed by β3 and β4. Additionally, hydrophilic zipper-like interactions formed by residues 61–65 and 79–80 contribute to the stabilization of the U-shaped structure (Supplementary Fig. 6d). β3 connects to the two U-shaped structures through hydrophobic zipper-like interactions with β2, mediated by residues 66-71 (Fig. 3a). β1 further wraps around the U-shaped structures, forming the iron-like core of S87 (Supplementary Fig. 6e). Intriguingly, despite that unmodified WT α-syn polymorph 1a employs the same region (residues 37-99) for forming fibril core[44], it features varying numbers of β-strand formed by different segments, which assemble into a unique Greek key-like structure (Figs. 3b, 4a, b). In the WT polymorph 1a, MSA fold and Juvenile-onset synucleinopathy (JOS) fold[44,48,49], residue S87 is not involved in direct interactions with other residues (Fig. 3b, Supplementary Fig. 8c, d). Conversely, in WT polymorphs 2a, 2b and Lewy fold[41,46], S87 engages in zipper-like interactions (Supplementary Figs. 7e, 8c, d). While, K80, which forms an interaction with GlcNAc at S87 in the gS87 fibril structure, establishes a salt bridge with E46 at the edge of the Greek key of unmodified WT polymorph 1a (Fig. 3b). This K80-E46 salt bridge is essential for maintaining the entire Greek key-like structure, and its disruption by the E46K mutation effectively abolishes the Greek key-like fold[45]. Therefore, the GlcNAc modified at S87 formed new interactions with K80 and E61, accompanied by the structural rearrangement of α-syn and the formation of a distinct β-strand pattern to create a distinct fibril core structure (Fig. 4a, b).

Remarkably, when phosphorylated, S87 is excluded from the fibril core, contrasting sharply with its O-glycosylated counterpart in the fibril structure (Fig. 3a). Starting from residue 73, the extended C-terminal region (residues 73-140) remains highly flexible in the fibril structure, without electron density. Conversely, the entire N-terminal region (residues 1-35), absent in both gS87 and unmodified α-syn fibril cores, is incorporated into the pS87 fibril core (Fig. 3c). In the pS87 fibril structure, residues 1-29 from the N-terminal form a U-shaped structure stabilized by two pairs of salt bridges, including K6 & E20 and K21 & D2 (Fig. 3c). Meanwhile, V3 and M5 on β1 establish hydrophobic interactions with residues L38' and V40' from the neighboring chain, forming intermolecular interactions that bring the two protofilaments together (Fig. 3c). The second U-shaped structure is formed by residues 30-49, stabilized by a salt bridge between K32 and E46 and a hydrogen bond between K34 and Y39 (Supplementary Fig. 6f). β2

(residues 26-29) forms a steric zipper-like hydrophobic interaction with residues 52-55, connecting the two U-shaped structures (Supplementary Fig. 6f).

Intriguingly, our prior structural results demonstrated that phosphorylation at Y39 of α-syn enables the integration of the entire N-terminal region into the fibril core[7] (Fig. 3d). In the pY39 fibril structure, the phosphate group modified at Y39 establishes extensive electrostatic interactions with K21, K32, and K34 (Fig. 3d). These interactions are vital for incorporating the entire N-terminal region into the fibril core, resulting in a large hook-like architecture (Fig. 4a, b). This structure suggests that if a phosphate group participates in fibril core formation, it needs to be stabilized by positively charged residues[7]. However, in the pS87 fibril structure, the positively charged residues are mostly localized in the N-terminal region, which either form electrostatic interactions with negatively charged residues (K6 & E20, K21 & D2, K32 & E46) or are situated on the fibril's outer surface (Fig. 3c). Consequently, the addition of a phosphate group to S87 prevents the C-terminal region of NAC from participating in fibril core formation due to electrostatic repulsion, obstructing α-syn's ability to form the Greek key-like fold observed in unmodified WT α-syn fibril. In contrast, the entire N-terminal region is incorporated into the fibril core, giving rise to a unique arch-like architecture in the pS87 fibril.

## gS87 and pS87 α-syn fibrils exhibit reduced neurotoxicity and propagation activity

Finally, we investigated whether the fibril structures induced by glycosylation and phosphorylation at S87 display altered neuropathological activities using a well-established neuronal propagation model[50–53] (Fig. 5a). We added unmodified WT, gS87, and pS87 α-syn PFFs into the culture medium of rat primary cortical neurons to seed the aggregation of endogenous α-syn. After 14 days of PFFs treatment, we detected the formation of the PFF-induced endogenous α-syn aggregate with an antibody of pathological pS129 α-syn through immunofluorescence staining. Remarkably, both groups treated with gS87 and pS87 α-syn PFFs displayed a significant reduction in pathological pS129 α-syn propagation compared to the unmodified WT α-syn PFFs-treated group (Fig. 5b, c). The pS87 PFFs exhibited the lowest propagation activity (Fig. 5c). Moreover, we measured the cytotoxicity of unmodified WT, gS87, and pS87 α-syn fibrils to neurons using a cell counting kit-8 (CCK-8) assay. The result showed that both gS87 and pS87 α-syn PFFs exhibited considerably decreased toxicity to primary neurons compared to unmodified WT α-syn PFFs, with pS87 PFFs displaying the lowest neurotoxicity (Fig. 5d). In conclusion, our findings indicate that glycosylation and phosphorylation at S87 induce α-syn to form two distinct fibril structures with reduced propagation activity and neurotoxicity in neurons.

## Discussion

Various types of PTMs have been identified that modify α-syn at different sites, playing diverse roles in regulating α-syn monomer conformation, membrane binding, amyloid aggregation kinetics and the atomic structures and pathology of α-syn fibril[7,29]. In this study, we explore whether distinct PTMs at the same residue can lead to the formation of different fibril structures. Intriguingly, we found that O-GlcNAcylation at the S87 residue can introduce new intra-molecular interactions, resulting in the formation of a iron-like fold in the fibril structure (Figs. 3a, 4a). This O-GlcNAcylation generates a distinct fibril polymorph by directly establishing new interactions. In contrast, phosphorylation at the same site prevents the involvement of the C-terminal region of NAC in the fibril core due to electrostatic repulsion. Instead, it promotes the incorporation of the entire N-terminal region to form a unique, enlarged arch-like fibril core structure (Figs. 3c, 4a). Therefore, this phosphorylation generates another fibril polymorph by excluding certain NAC region from forming fibril core.

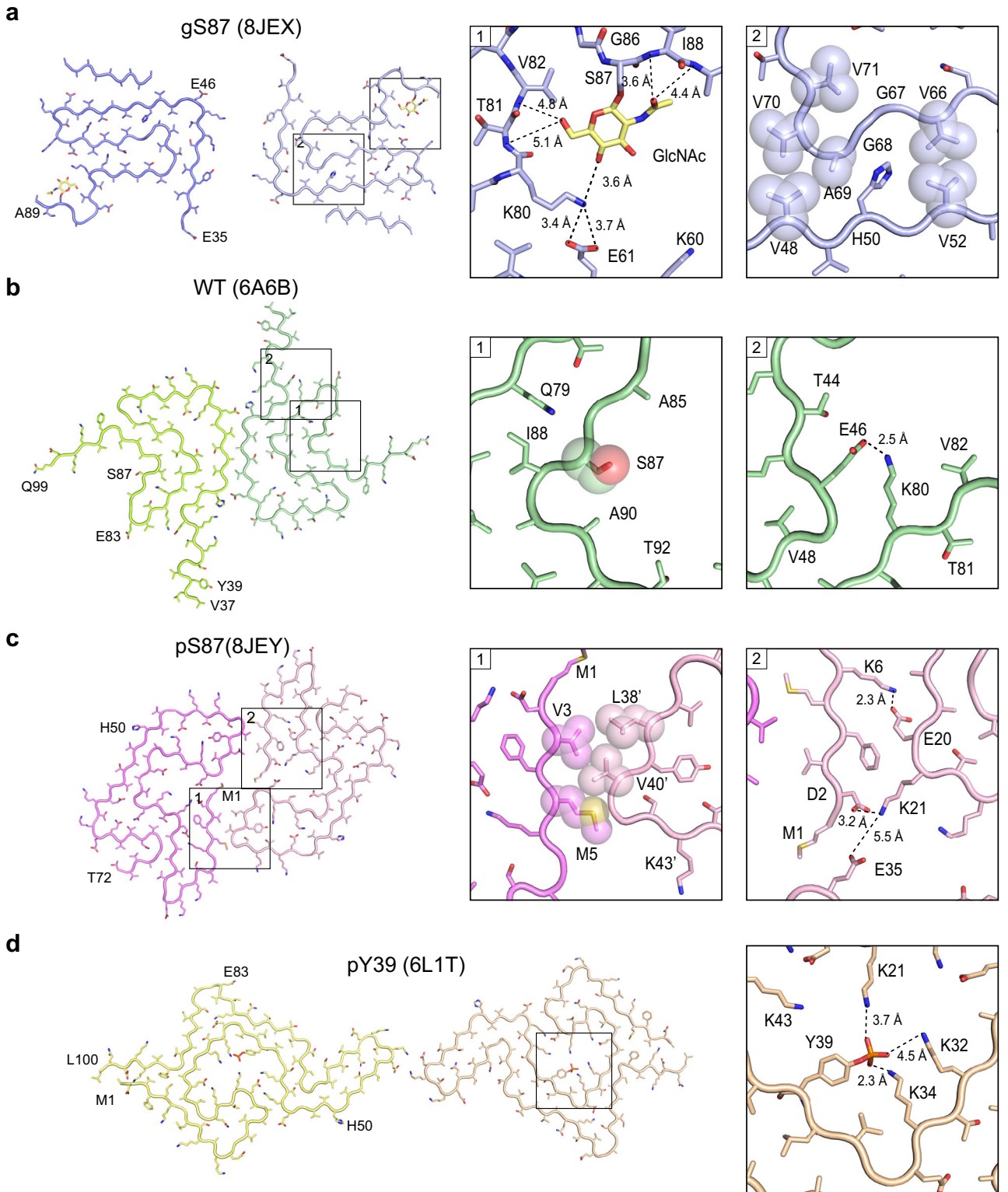

**Fig. 3 | Structural analysis of gS87, pS87, unmodified WT, and pY39 α-syn fibril.**
**a** The structural model of gS87 fibril, with the zoom-in views the interactions between GlcNAc, K80, E61, T81, V82, I88 and A89, and the hydrophobic zipper-like interactions with the involved residues labeled. **b** The structure of unmodified WT α-syn fibril, with the conformation of S87 and the salt bridge between K80 and E46 shown in the zoom-in views. **c** The structure of pS87 fibril, with the hydrophobic interactions of the interface between two protofilaments, and two pairs of salt bridges shown in the zoom-in views. Residues involved in the inter-protofilamental interactions are shown in spheres. **d** The structure of pY39 fibril with the electrostatic interactions of the phosphate group bound to Y39 and K21, K32 and K34 shown in the zoom-in view. Distances are shown in Å. The PDB code of each fibril structure is indicated.

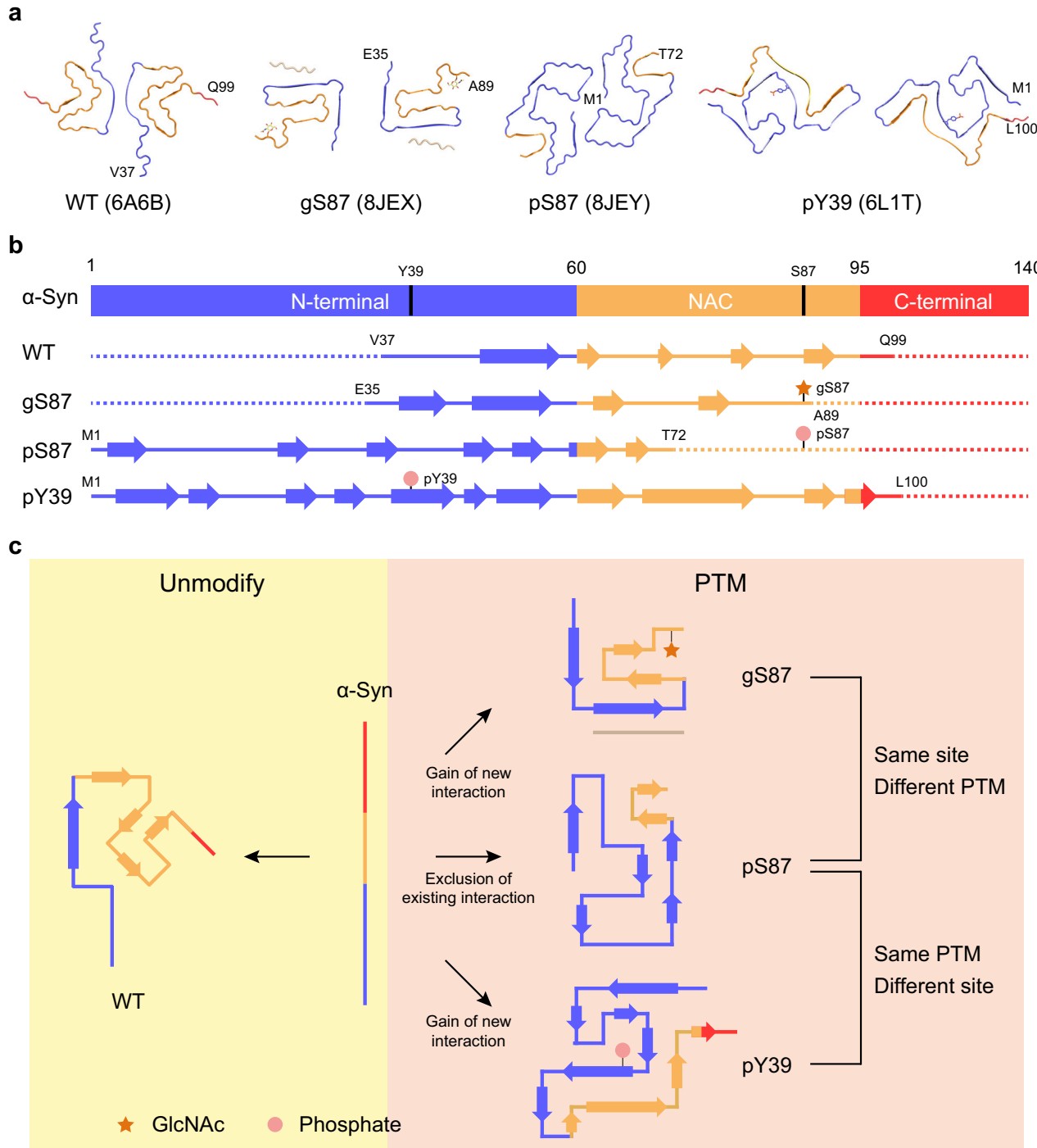

**Fig. 4 | Comparison of α-syn fibril structures with different modifications.**
**a** one-layer models of unmodified WT (PDB: 6A6B), gS87 (PDB: 8JEX), pS87 (PDB: 8JEY) and pY39 (PDB: 6L1T) fibrils with the N-terminal region colored in blue, the NAC colored in yellow and the C-terminal region colored in red. **b** The secondary structure alignment of four α-syn fibril structures from (**a**) with different colors for three regions. **c** Schematic diagram shows that both same PTM modified at different sites and different PTMs modified at same site induce distinct fibril core structures.

Both phosphorylation and O-GlcNAcylation at S87 generate fibril structures distinct from structures of in vitro WT fibrils[44,46,54] (Supplementary Fig. 7) and ex vivo fibril[41,48,49] (Supplementary Fig. 8). The different conformations of gS87 and pS87 in comparison to ex vivo fibrils indicate that these modified forms alone do not replicate the conformations of fibrils extracted from patient brains. It is important to note that the fibrils found in the brain are heterogeneous, encompassing a variety of PTMs. Current ex vivo structures may represent the end stage species, and potentially overlook certain minor species. Both O-GlcNAcylation and phosphorylation at S87 mitigate α-syn aggregation, potentially leading to the formation of fibrillar species that do not represent the terminal state in disease contexts. Future investigations focusing on mixed PTMs and the identification of intermediate species during disease progression could yield insights into the impact of PTM cross-talk on α-syn aggregation in real disease conditions. Consequently, different PTMs at the same site may utilize entirely distinct mechanisms to direct fibril core formation, depending on the specific properties of the modified groups (Fig. 4c).

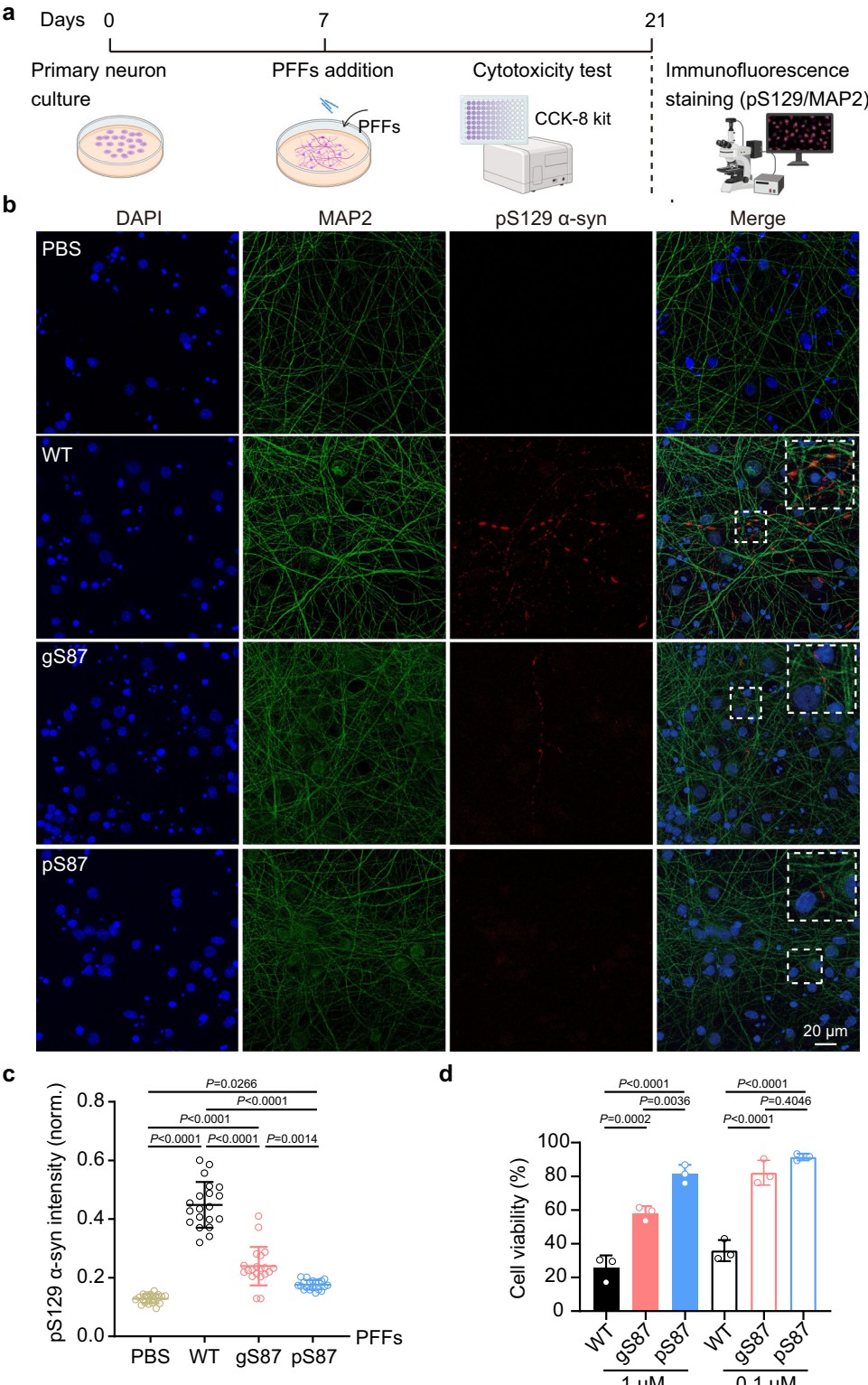

**Fig. 5 | Neurotoxicity and propagation activity measurement of the unmodified WT, gS87 and pS87 α-syn fibrils. a** Workflow of cytotoxicity test and immunofluorescence staining in rat primary neurons. The workflow is created with BioRender.com. **b** Representative immunofluorescence images of rat primary neurons treated with PBS, 100 nM unmodified WT PFFs, gS87 PFFs and pS87 PFFs for 14 d, respectively. DAPI (blue), pS129 α-syn (red) and microtubule-associated protein 2 (MAP2) (green) were stained. Scale bar: 20 μm. Images were processed by Image J. **c** Quantitative analyses of pS129 α-syn aggregation induced by different types of fibrils. The intensity of pS129 α-syn was normalized to the intensity of DAPI for each sample. One-way ANOVA followed by Tukey's post-hoc test. Data shown are mean ± s.d., $n = 20$, 6–8 images were randomly taken for each sample. 3 individual samples were analyzed for each group. **d** Cytotoxicity measurement of the unmodified WT, gS87 and pS87 PFFs using CCK-8 kit. One-way ANOVA followed by Tukey's post-hoc test. Data shown are mean ± s.d., $n = 3$ independent samples (biological repetitions). Source data are provided as a Source Data file.

Additionally, we compared the fibril structures of pS87 α-syn with pY39 α-syn, and found that the same phosphate group installation at different sites can also produce completely different fibril structures (Figs. 3c, d, 4a). In pY39 α-syn, the phosphate group added at Y39 forms extensive interactions with three lysine residues in the N-terminal region, inducing the formation of a large hook-like fibril structure (Figs. 3d, 4a). However, adding the phosphate group at S87 results in the exclusion of the C-terminal region of NAC from incorporation into the fibril core due to electrostatic repulsion. Thus, the local context of the modified site may directly impact the regulation mechanism of a given modified group, adding an additional layer of complexity to the regulation of fibril structural polymorphs by PTMs (Fig. 4c). Of note, cryo-EM structure captures α-syn fibrils in their final aggregated state. Consequently, it remains unclear whether PTM-mediated interactions occur prior to or concurrent with the formation of the rest of the fibril cores. To elucidate this, further studies are required to characterize the kinetic intermediates of protein fibrillar assembly at the atomic level.

Notably, both gS87 and pS87 α-syn fibrils display considerably lower neurotoxicity and propagation activity compared to the unmodified WT α-syn fibril (Fig. 5b–d). This finding aligns with previous observations that both pS87 and gS87 modifications protect against α-syn aggregation and pathology in cells[8,15]. In stark contrast, the pY39 modification exacerbates α-syn pathology in cells and produces a fibril polymorph with heightened neurotoxicity[7]. However, the direct relationship between the structural polymorphs and their varying neuropathology is not yet well-established. Further investigation is necessary to understand how gS87 and pS87 fibril structures lead to reduced neuropathology. Furthermore, since both PTMs share the same modification site on α-syn, exploring the cross-talk and potential competition between these two modifications at S87 during various physiological or pathological processes, and its implication in disease progression, is of importance.

## Methods

### Ethical statement
Embryonic day 16–18 Sprague-Dawley rats embryos used in this paper were purchased from Shanghai SIPPR BK Laboratory Animals Ltd, China. All animal and cell experiments in this study were performed following the protocols approved by the Animal Care Committee of the Interdisciplinary Research Center on Biology and Chemistry (IRCBC), Chinese Academy of Sciences (CAS).

### Semi-synthesis of gS87 and pS87 α-syn
The native chemical ligation strategy using peptide hydrazides of Liu and co-workers[34,35] was used to synthesize α-syn with PTMs. We performed N-to-C sequential native chemical ligation using three segments referring to the work of Pratt and co-workers[8,32,36], including segment X (α-syn 1-84 thioester), segment Y (pS87/gS87 α-syn A85C-90NHNH$_2$), segment Z (α-syn A91C-140) (Fig. 1a).

To obtain segment X, α-syn 1-84 was introduced into pTWIN1 vector (NEB) containing intein and chitin binding domain. BL21(DE3) *E. coli* was transformed and then cultured in Luria-Bertani (LB) broth with 50 μg/mL ampicillin sodium salt. Protein expression was induced by addition of 1 mM isopropyl-β-D-1-thiogalactopyranoside (IPTG) and incubation of cells for 20 h at 16 °C. Then, the cells were harvested in buffer (50 mM Hepes, 500 mM NaCl, pH 7.5) and lysed by ultrasonication (twice on ice, each for 20 min). The lysate was centrifuged ($21,500 \times g$, 90 min), and the supernatant was loaded on chitin column (NEB). After cleavage in 250 mM mercaptoethanesulfonate (MesNa), the protein thioester was concentrated by ultrafiltration (Supplementary Fig. 2a).

Segment α-syn Y-pS87 (CG[pS]IAA-NHNH$_2$) and Segment α-syn Y-gS87 (CG[gS]IAA-NHNH$_2$) were manually synthesized using 2-Cl-(Trt)-NHNH$_2$ resin as described[34]. Fmoc-L-Ser (GlcNAc(Ac)$_3$-β-D)-OH was synthesized referring to the work of Pratt[37] and characterized with RP-HPLC, ESI-MS, and NMR (Supplementary Fig. 1). Fmoc-L-Ser(GlcNAc(Ac)$_3$-β-D)-OH was coupled on peptide using 1.8eq HATU, 2eq HOAT, 5eq N-methylmorpholine (NMM) in N-Methylpyrrolidone (NMP). And the deprotection of O-acetyl groups on GlcNAc was carried out using 60% hydrazine hydrate in CH$_3$OH (v/v) on resin for 15 min. Reagent K was used in cleavage. The peptides were purified and characterized with RP-HPLC (YMC-Pack ODS-A column) and ESI-MS (Supplementary Fig. 2b, c).

For segment Z, BL21(DE3) *E. coli* was transformed with pET-22b vector containing α-syn A91C-140. After incubation and expression. The protein was purified using osmotic-shock strategy[55]. Briefly, after centrifuging, the collected cells were treated with osmotic shock buffer (30 mM Tris-HCl, 40% sucrose (w/v), and 2 mM EDTA, pH 7.4). After centrifugation ($15,700 \times g$, 20 min), cells were suspended in cold water with saturated MgCl$_2$ added. Then the supernatant was collected. Extra precipitation (with HCl to pH 3.5 and NaOH to pH 7 in sequence) was also applied to increase purity. 400 mM O-methylhydroxylamine and 20 mM Tris(2-carboxyenthyl)phosphine (TCEP) was then added and incubated for 5 h to reverse N-terminal cysteine modification[38]. α-syn A91C-140 was further purified using RP-HPLC with proteonavi column (OSAKA SODA), then lyophilized for next reaction.

Then we ligated X and Y. Briefly, X thioester (2.5 mM, 1 eq.) and Y-pS87/Y-gS87 (2 eq.) were dissolved in ligation buffer (6 M guanidine-HCl, 200 mM phosphate buffer, pH 7.0). Then, 50 eq. TCEP and 4-mercaptophenylacetic acid (MPAA) (in 200 mM phosphate buffer, pH 6.8) were added. The solution was shacked at 30 °C for 5 h, then purified using RP-HPLC with proteonavi column. The product XY-pS87/gS87 was lyophilized for next reaction (Supplementary Fig. 2d).

For the ligation of XY and Z, segment XY (5 mM, 1eq.) was dissolved in 6 M guanidine-HCl, 200 mM phosphate buffer at pH 3.0 and put into a −15 - −20 °C ice−salt bath. 0.5 M NaNO$_2$ (10 eq.) was added in the reaction or 15 min. After that, 40 eq. MPAA and 1.2 eq. segment Z in buffer (200 mM phosphate buffer, 6 M guanidine-HCl, pH 6.5) was added and then the pH was adjusted to 6.8. The final concentration of XY is about 2.5 mM. The mixture was shaken at 16 °C overnight and then purified using RP-HPLC with proteonavi column. The product XYZ-pS87/gS87 was lyophilized for next reaction. The segments after ligation were all characterized with analytical RP-HPLC and ESI-MS (Supplementary Fig. 3).

Radical catalyzed desulfurization was then performed to obtain α-syn protein with different modifications at serine 87. Briefly, about 2.6 mg of XYZ- pS87/gS87 was dissolve in 200 μl of 6 M guanidine-HCl, 200 mM sodium phosphate buffer at pH 7.0 and mixed with 200 μl of 1 M TCEP and 40 μl of 2-methyl-2-propanethiol. 20 μl of 0.1 M 2-2'-azobis[2-(2-imidazolin-2-yl)propane] dihydrochloride (VA-044) was finally added to the mixture under argon. The reaction was shaken overnight at 37 °C. The pS87/gS87 α-syn protein was purified with proteonavi column, lyophilized, and characterized with analytical RP-HPLC and ESI-MS (Supplementary Fig. 4).

### Mass spectrometry
The mass spectrometry data were collected using Chameleon 7.2.10. on Thermo Scientific™ MSQ Plus mass detector.

The MSQ Plus mass spectrometry settings: Needle (capillary) voltage: 3.0 kV. Cone voltage: 15 V for Y-gS87, 45 V for other samples. Mobile phase: 0.4 mL/min 50% CH$_3$CN/H$_2$O with 0.06% formic acid. Probe temperature: 350 °C. Number of replicates for each peptide or protein sample is $n = 1$.

GraphPad Prism 9 was applied for graphing.

### Nuclear magnetic resonance spectroscopy
The $^1$H-NMR and $^{13}$C-NMR data of Fmoc-L-Ser(GlcNAc(Ac)$_3$-β-D)-OH were both collected using Delta v4.3.3 on JEOL ECS 400.

Settings for $^1$H-NMR:

Temperature: 22.1 °C. single pulse. Pulse width: 5.2500. Relaxation delay: 5.0000. Number of scans: 12. Receiver gain: 40.

Settings for $^{13}$C-NMR:

Temperature: 21.9 °C. single pulse. Pulse width: 2.5833. Relaxation delay: 2.0000. Number of scans: 1000. Receiver gain: 54.

Mestnova v9.0.1 was applied for data processing.

## Preparation of the unmodified WT, gS87 and pS87 α-syn fibrils and PFFs

Recombinant unmodified WT, synthetic gS87 and pS87 α-syn in buffer containing 25 mM sodium phosphate (pH 7.4) were incubated at 37 °C, 900 rpm in ThermoMixer (Eppendorf) for 4 days with agitation in a concentration of 3.0 mg/ml, respectively. The fibril samples were further used for negative staining transmission electron microscopy, atomic force microscopy, cryo-EM sample preparation and rat primary neuron treatment. For incubated fibrils of unmodified WT, gS87 and pS87 α-syn, the fibril-containing pellet is separated from the monomer-containing supernatant by centrifugation. Then, the fibril-containing pellet was resuspended and its concentration was determined as the fibril concentration. Unmodified WT, gS87 and pS87 α-syn PFFs were prepared by fibrils sonication at 20% power on ice for 25 times (1 s on, 1 s off).

## ThT kinetic assay

50 μM unmodified WT, gS87 and pS87 α-syn monomers were incubated in 25 mM sodium phosphate (pH 7.4) buffer with or without unmodified WT α-syn PFFs in a black 384-well plate (Thermo Scientific). ThT was added to the reaction mixture at a final concentration of 30 μM. The fluorescence intensity was recorded using a Varioskan Flash Spectral Scanning Multimode Reader (Thermo Scientific), measuring at 440 nm (excitation) and 485 nm (emission) wavelengths with shaking at 900 rpm at 37 °C. Three replicates were performed for each sample. GraphPad Prism 9 was applied for graphing, with mean ± s.d.

## Negative staining transmission electron microscopy

Five microliters of fibril solution were incubated on a 200-mesh glow-discharged copper grid (Zhongjingkeyi Technology Co., Ltd., Beijing) for 45 s. Then, the grid was washed with double-distilled water followed by 2% w/v uranyl acetate for another 45 s, and dried in air. The samples were imaged by a Tecnai T12 transmission electron microscope (FEI) operated in 120 kV.

## Atomic force microscopy

Ten microliters of fibril solution were loaded on a clean mica surface for 5 min at room temperature and washed by double-distilled water into remove unbound fibrils. Next, images were captured by Nanoscope V Multimode 8 (Bruker) with SNL-10 probes (a constant of 0.35 N m$^{-1}$) on ScanAsyst air mode in 1 Hz scan rate. The following data processes were carried out in the supplied software NanoScope Analysis (version 1.5, Bruker).

## Cryo-EM sample preparation and data collection

The gS87 and pS87 α-syn fibril samples were applied into glow-discharged holey carbon copper grids (R2/1, 300 mesh, Quantifoil), and then plunge frozen in liquid ethane after blotting with filter paper by using Vitrobot Mark IV (Thermo Fisher).

Cryo-EM data of gS87 and pS87 was collected in a 300 kV Titan Krios G4 transmission electron microscope (Thermo Fisher) with a BioContinuum K3 direct detector (Gatan), using a GIF Quantum energy filter (Gatan) with a slit width of 20 e·V to remove inelastically scattered electrons. Movies with 40 frames per micrograph were recorded in ×105,000 magnification with a pixel size 0.83 Å pixel$^{-1}$ at super-resolution mode. Cryo-EM data collection was performed by software EPU (Thermo Fisher) with 2 s exposure time and −1.0 to −2.0 μm defocus value in a total dose of 55 e$^-$Å$^2$.

## Image processing

MotionCorr2[56] for motion correction implement was carried out to correct beam-induced motion of movie frames with dose-weighting. Then, CTFFIND-4.1.8[57] was used to estimate the contrast transfer function of motion-corrected images. Next, fibrils were manually picked by using the manual picking method of RELION version 3.1[58].

## Helical reconstruction

Helical reconstruction was performed in RELION version 3.1[58], including particle extraction, two-dimensional (2D) classification, 3D classification, 3D auto-refinement and post-processing.

For gS87 dataset, 21,328 manually fibrils from 2134 micrographs were extracted into segments in a box-size of 360 pixels with an inter-box distance of 30.0 Å. Then, several iterations of 2D classification were applied to extracted segments with a decreasing in-plane angular sampling rate from 12° to 0.5° and the T = 2 regularization parameter. For double filament and single filament gS87 α-syn fibrils separated after 2D classification, a cylindrical map model generated by relion_helix_toolbox program was used as an initial 3D reference to perform the following 3D classification (K = 3). When the local search of symmetry for helical twist and rise was carried out after the separation of β-strands, the clearest class was selected for following 3D auto-refinement. Finally, the overall resolution of double filament gS87 α-syn fibrils was reported at 3.1 Å, according to the gold-standard Fourier shell correlation (FSC) = 0.143 criteria.

For pS87 dataset, 27,806 fibrils from 2423 micrographs were extracted into segments in a box-size of 360 pixels with an inter-box distance of 30.0 Å. Next, several iterations of 2D classification were applied to extracted segments with a decreasing in-plane angular sampling rate from 12° to 0.5° and the T = 2 regularization parameter. Similarly, for the twisted filament of pS87 α-syn fibril separated from 2D classification, a cylindrical map model generated by relion_helix_toolbox program was used as an initial 3D reference to perform the following 3D classification (K = 3). When the local search of symmetry for helical twist and rise was carried out after the separation of β-strands, the clearest class was selected for following 3D auto-refinement. Finally, the overall resolution of twisted filament gS87 α-syn fibrils was reported at 2.6 Å, according to the gold-standard FSC = 0.143 criteria.

## Atomic model building and refinement

Based on the density maps after post-processing program, the atomic models of double filament of gS87 α-syn fibril and twisted filament pS87 α-syn fibril were built de novo in COOT[59], respectively. Then, three-layer models were generated in software Chimera and refined by real-space refinement program of PHENIX[60,61]. Additional details about two models were shown in Supplementary Table 1.

## Primary neuronal cultures

Embryonic day (E) 16 to E18 Sprague-Dawley rat (Shanghai SIPPR BK Laboratory Animals Ltd.) embryos were sacrificed for primary cortical neurons culture as previously described[62]. In brief, papain-digested cortical neurons were plated onto coverslips, coated with poly-L-lysine, in a 24-well plate at a density of 150,000 cells per well. After 7 days, neurons were treated with PBS, 100 nM WT or gS87 or pS87 α-syn PFFs, with three biological repetitions for each treatment. All animal experiments were performed according to the protocols approved by the Animal Care Committee of the Interdisciplinary Research Center on Biology and Chemistry (IRCBC), Chinese Academy of Sciences (CAS).

## Immunofluorescence staining and confocal imaging

After treating for 14 days, neurons were collected for immunofluorescence imaging. Neurons plated on coverslips were fixed with 4% paraformaldehyde (PFA) and 4% sucrose in PBS and then

permeabilized with 0.15% Triton X-100 diluted in PBS. Then coverslips were blocked with 3% goat serum (GS) diluted in PBS for 30 min at room temperature. Neurons were incubated with primary antibodies of phospho-α-synuclein (1:1000, Abcam, 51253) and MAP2 (1:2500, Abcam, 5392), diluted in 3% GS, at 4 °C overnight. After washing with PBST, 0.1 % Tween-20 diluted in PBS for three times, neurons were incubated with secondary antibodies of Alexa Fluor 488- and Alexa Fluor 568- (1:1000, Invitrogen, 2420700, 2155282, respectively) for 1 h at room temperature. Without rinsing, DAPI stain was applied (1:10,000, Yeasen, 40728ES03) for 10 min at room temperature. After rinsing, coverslips were mounted onto glass slides with mounting medium (ProLong Gold antifade reagent, Invitrogen, P36930). Confocal images were acquired by a laser scanning confocal microscope (SP8, Leica). A 63× water immersion objective was used for imaging. Images were batched analyzed by Image J. All images were projected in z direction with max intensity followed by background subtraction. The immunofluorescence results of the mean p-α-syn signal intensity were normalized to DAPI intensity of their own group.

## Cell viability assay

Primary neurons were treated with different concentrations of unmodified WT, gS87 and pS87 α-syn PFFs or PBS after growing for 7 d. Three biological replicates were set up for each treatment. After treating for 14 d, the CCK-8 assay was performed following the manufacturer's instructions. The absorbance at 450 nm wavelength was measured by a multimode plate reader (Ensight, PerkinElmer). Values were analyzed by GraphPad Prism 9.

## Statistics

At least three independent biological repeats were performed for each group. Sample sizes were selected to ensure sufficient statistical power. Data of pS129 α-syn aggregation and cytotoxicity measurement were analyzed using One-way ANOVA followed by Tukey's post-hoc test. Data shown are mean ± s.d. A $p < 0.05$ was considered to be statistically significant. Detailed statistical information is shown in figure legends.

## Reporting summary

Further information on research design is available in the Nature Portfolio Reporting Summary linked to this article.

## Data availability

Density maps of double filament of gS87 α-syn fibril and twisted filament pS87 α-syn fibril have been deposited in Electron Microscopy Data Bank (EMDB) under accession codes: EMD-36202 for double filament of gS87 α-syn fibril and EMD-36203 for twisted filament pS87 α-syn fibril. And, the structure models have been deposited in the Protein Data Bank (PDB) with entry codes: 8JEX for double filament of gS87 α-syn fibril and 8JEY for twisted filament pS87 α-syn fibril. The PDB codes for WT$_{1a}$, WT$_{2b}$, WT$_{2a}$, WT$_{2b}$, pY39, Lewy fold, MSA fold, and JOS fold are 6A6B, 6CU8, 6SSX, 6SST, 6L1T, 8A9L, 6XYO, and 8BQV, respectively. All data needed to evaluate the findings of this study are available in the article and Supplementary files. Source data are provided with this paper.

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

## Acknowledgements

We thank the Cryo-EM Microscopy Center at Interdisciplinary Research Center on Biology and Chemistry, Shanghai Institute of Organic Chemistry for help with data collection. This work was supported by the National Natural Science Foundation of China (Grant No. 92353302 to Y.-M.L., 92053108 to Y.-M.L., 82188101 to C.L., 32170683 to D.L. and 32171236 to C.L.,), the National Key R & D Program of China (2019YFA0904200 to Y.-M.L., 2018YFA0507600 to Y.-M.L., 2019YFE0120600 to C.L.), the Science and Technology Commission of Shanghai Municipality (STCSM) (Grant No. 20XD1425000 to C.L., 2019SHZDZX02 to C.L. and 22JC1410400 to C.L.), the Shanghai Pilot Program for Basic Research – Chinese Academy of Science, Shanghai Branch (Grant No. JCYJ-SHFY-2022-005 to C.L.), the CAS Project for Young Scientists in Basic Research (Grant No.YSBR-095 to S-N.Z.).

## Author contributions

C.L. and Y-M.L. designed the project. J-J.H. and Y-J.L. synthesized the modified protein. L.Z. synthesized the glycosylated Fmoc-amino acid. H-S.W. is involved in the synthesis of modified protein. Y-P.S., Y-Q.T. and W-C.X. performed the sample optimization. W-C.X. collected and processed the cryo-EM data. S-Y.Z. and W-D.L. performed the ThT kinetic assay, propagation and cytotoxicity assays, and analyzed the data. C.L., Y-M.L., S-N.Z. and D.L. wrote the manuscript. All the authors are involved in analyzing the data, and contributed to manuscript discussion and editing.

## Competing interests

The authors declare no competing interests.
