## [Peer Review File · Nature Communications]

Phosphorylation and O-GlcNAcylation at the same α -synuclein site generate distinct fibril structuresReviewer #1 (Remarks to the Author):

This manuscript by Hu et al. reports the cryo-EM structures of α -synuclein fibrils with phosphorylation (pS87) or O-glcNAcylation (gS87) at residue S87. The proteins of pS87 and gS87 were prepared via a combination of chemical synthesis and recombinant expression. Aggregation propensity of pS87 and gS87 was studied using chemical kinetics and toxicity of pS87 and gS87 fibrils was studied using cell-based toxicity assays. The authors concluded that phosphorylation and O-GlcNAcylation reduced aggregation propensity, led to the formation of different fibril structures, and lowered the fibril-induced toxicity. The structural work adds to a growing body of α -synuclein fibril structures and is a worthy contribution to the α -synuclein field. However, at the current form, the manuscript lacks some clarity on several key questions and can benefit from an expanded discussion of their own results and other related studies.

1. The gS87 fibrils have two fibril polymorphs: double filament and single filament. The pS87 also has two polymorphs: straight and twisted. The structure of only one of the two fibril polymorphs was solved using cryo-EM. In the case of pS87, the structure of the twisted fibril polymorph was determined, but it is the minor form (26%). Is it possible that the unsolved fibril polymorph represents a similar structure to wild-type fibrils? Furthermore, it is well-known that α -synuclein fibrils adopt a variety of fibril polymorphs. In the text, only one WT fibril structure was discussed. How do the gS87 and pS87 fibril structures compare with other WT fibril structures? It is more nuanced than the simple conclusion to say the PTMs at S87 led to the formation of different structures.

2. Methods for the toxicity assay lack some key details. In the figure legend of Figure 5, it appears that WT, pS87 and gS87 fibrils are at 100 μ M concentration. Is this the monomer equivalent concentration? If so, are there any purification steps performed after fibril preparation? Are the amounts of fibrils normalized for different fibrils? The authors have shown that gS87 and pS87 have lower aggregation propensity than WT, so gS87 and pS87 at the same monomer concentration will have lower amounts of fibrils than WT. Then the difference in toxicity could be caused by the amount of fibrils rather than the structure of the fibrils. Furthermore, there are different fibril polymorphs for WT, gS87, and pS87 fibrils. Have the authors attempted to purify the specific fibril polymorph? If not, how does the presence of different fibril polymorph factor in the interpretation of results?

3. The cryo-EM structures show ordered and disordered regions. Can the authors determine the degree of disorder for the unresolved segments? It would be helpful to determine whether the disordered regions are just barely disordered for cryo-EM or completely disordered. In future studies, it would be helpful to characterize the disordered regions with NMR or EPR to get a sense of disorderedness. In the gS87 structure, the authors identified an ordered island, but stopped short of revealing the identity of these residues. Could these residues be connected to residue 89 via a disordered region?

4. All the fibrils were prepared in vitro. It is important to note the limitations of in vitro fibrils as many studies have shown that fibrils isolated/purified from brain tissues adopt different structures from those prepared in the test tubes.

Some minor points:

1. Figure 1b. TEM picture of seeded gS87 fibrils shows significantly larger fibril width than the wild-type. The scaling should be checked to make sure the gS87 and WT fibrils are shown at the same magnification.

2. Figure 3. Suggest to flip the WT fibril structure (Figure 3b) vertically, so that the orientation resembles gS87. In the flipped orientation, it is more obvious that the WT fibril and gS87 have an overall resemblance at the backbone level even though the detailed packing is different.

3. ThT was never spelled out.

Reviewer #2 (Remarks to the Author):

This study looks at α -syn fibrils with phosphorylation and O-GlcNAcylation at the site S87. This produces distinct different fibril morphologies than the unmodified protein and also reduces neurotoxicity and propagation of fibril compared to unmodified protein. This helps the further understanding of posttranslational modifications in amyloid diseases.

Overall, I support the publication of the manuscript. The methodology is sound and meets the expected standards.

But I have some comments and questions for the authors:

How would the modification at position S87 impact the structures of ex vivo α -syn fibrils (Lewy fold, MSA folds)? A discussion or comparison to the ex vivo fibril structures in relation to your findings would be appreciated.

Minor comments and questions:

A proteolysis experiment of unmodified and modified α -syn fibrils could be interesting, because stability against proteases seems to be a major factor for which morphologies are found in vivo.

Did you do polishing in Relion after 3D auto-refinement?

One or two decimal points for the B-factor in Supplementary Table 1 are enough.

Clash score is pretty high for the estimated resolution. It should be possible to get below 5.

Reviewer #3 (Remarks to the Author):

The manuscript by Hu et al describes structural studies of two post-translationally modified alpha synuclein fibrils at a single site, S87, and their pathologically relevant effects on the propagation of amyloid aggregation and neurotoxicity in primary neurons. Chemical ligation and synthesis yielded high purity alpha synuclein proteins with single site modification, which allowed studies of site-specific effects. Using cryo-TEM 3D reconstruction, both fibril structures were obtained at ~ 3 angstrom resolution, where distinct features between these PTM variants, and between PTM and wt alpha synuclein fibrils were identified. Potentially, the work may prompt the current understanding of the influence of site-specific PTM of amyloid proteins on their pathological roles. In the current version, some of the experiments/interpretation of results may be improved:

1. Biophysical characterization of fibrils.

(1) ThT-fluorescence: Please provide the kinetics traces for gS87 and pS87 fibrillation that show clear increment of fluorescence emission, especially for pS87.

(2) Negatively stained TEM: Please provide TEM images that shows fibrils morphologies for gS87 and pS87, especially for pS87. It is unclear from the current images whether or not there are fibrils. It is expected from the text that both fibrils should have twisted morphologies on their filaments.

(3) AFM: For gS87 and pS87, it seems that the pitches are heterogenous, are there any quantitative or semi-quantitative analysis of the pitch lengths based on the AFM? And how would these analyses related to the cryo-TEM structures?

2. Cryo-TEM structures.

(1) There is a large gap in its quaternary interface for gS87(the authors have also mentioned this in line 162-166, about 15 angstrom), without any identified electron density. What may be the interactions to hold this interface?

(2) It is mentioned that for pS87 fibrils, 74% of the fibrils have straight polymorph. Why was the 3D reconstruction done for the minor 26% twisted polymorph?

(3) It is stated in line 197-199: GlcNAc at S87 introduces new interactions with K80 and E61,

leading to a structural rearrangement.... What are the evidence that the structural difference between gS87 and wt fibrils are actually driven by this site-specific PTM? In other words, do the interactions between GlcNAc-S87 and the surrounding residues occur prior to the formation of the rest of fibrillar core?

(4) The discussion of pS87 effect on fibril structures in line 224-232 is not convincing because Fig. 3b shows that in wt alpha synuclein fibril, S87 is not involved in any tertiary or quaternary interactions. Why does this site-specific PTM have such a significant impact on the structure?

3. Immunostaining.

What is the justification for using the pS129-tau-specific antibody, but not total tau antibody, or both?

4. Minor points

(1) Line 50: Please give the full name of MSA.

(2) Fig. 2b: Should the helical rise be 2.41 angstrom instead of 4.82 for pS87?

Point-by-point response to the reviewers' comments:

* The reviewers' comments are in *Italic*. Author's responses are in blue. All revisions are colour highlighted in the revised manuscripts.

REVIEWER COMMENTS

Reviewer #1 (Remarks to the Author):

This manuscript by Hu et al. reports the cryo-EM structures of α -synuclein fibrils with phosphorylation (pS87) or O-glcNAcylation (gS87) at residue S87. The proteins of pS87 and gS87 were prepared via a combination of chemical synthesis and recombinant expression. Aggregation propensity of pS87 and gS87 was studied using chemical kinetics and toxicity of pS87 and gS87 fibrils was studied using cell-based toxicity assays. The authors concluded that phosphorylation and O-GlcNAcylation reduced aggregation propensity, led to the formation of different fibril structures, and lowered the fibril-induced toxicity. The structural work adds to a growing body of α -synuclein fibril structures and is a worthy contribution to the α -synuclein field. However, at the current form, the manuscript lacks some clarity on several key questions and can benefit from an expanded discussion of their own results and other related studies.

REPLY: We sincerely thank the reviewer for recognizing the importance and significance of our work. We also appreciate very much the insightful comments and suggestions provided by this reviewer. And we carefully addressed the reviewer's comments below and revised the manuscript accordingly.

1. The gS87 fibrils have two fibril polymorphs: double filament and single filament. The pS87 also has two polymorphs: straight and twisted. The structure of only one of the two fibril polymorphs was solved using cryo-EM. In the case of pS87, the structure of the twisted fibril polymorph was determined, but it is the minor form (26%). Is it possible that the unsolved fibril polymorph represents a similar structure to wild-type fibrils? Furthermore, it is well-known that α -synuclein fibrils adopt a variety of fibril polymorphs. In the text, only one WT fibril structure was discussed. How do the gS87 and pS87 fibril structures compare with other WT fibril structures? It is more nuanced than the simple conclusion to say the PTMs at S87 led to the formation of different structures.

REPLY: We deeply appreciate the valuable suggestion from the reviewer. Unfortunately, due to the technical limitations of cryo-EM, it's impossible to determine the structure of straight fibrils so far^{1, 2, 3}. Consequently, only the structure of the twisted pS87 fibril polymorph was determined. Since the major form of pS87 was straight fibrils, we don't know whether the unsolved fibril polymorph represents a similar atomic structure to wild-

type (WT) fibrils. Of note, the WT fibril was twisted and there is a clear difference between the 2D classification results of pS87 straight polymorph and WT fibril⁴ (Fig. R1-1a, b).

Following the reviewer's insightful suggestion, we compared gS87 and pS87 fibril structures with previously reported WT fibril structures^{5,6} (Fig. R1-1c-e), in addition to the WT_{1a} structure which was discussed in the previous manuscript (Fig. 3). The structural comparison among different α -syn fibril polymorphs shows that S87 doesn't participate in forming the fibril core in WT_{1b} structure, while it forms a zipper-like interaction with other residues in the WT_{2a} and WT_{2b} structures (Fig. R1-1e).

In the revised manuscript, we added Fig. R1-1 as Supplementary Fig. 7, and described it in Results, section "*Cryo-EM structure determination of gS87 and pS87 α -syn fibrils*" on page 6, as following:

".....Although the structural elucidation of straight fibrils remains challenging due to the current limitations of cryo-EM helical reconstruction technique^{41, 42, 43}, we observed a distinct difference in the 2D class averages between the pS87 straight filament polymorph and the WT polymorph 1a (Supplementary Fig. 7a, b). Besides, we....."

As well as in Results, section "*Comparison of α -syn fibril structures with two different modifications*" on page 8, as following:

".....Intriguingly, despite that unmodified WT α -syn polymorph 1a..... In the WT polymorph 1a, MSA fold and Juvenile-onset synucleinopathy (JOS) fold^{44, 48, 49}, residue S87 is not involved in direct interactions with other residues (Fig. 3b, Supplementary Fig. 8c, d). Conversely, in WT polymorphs 2a, 2b and Lewy fold^{41, 50}, S87 engages in zipper-like interactions (Supplementary Fig. 7e, 8c, d)..... of unmodified WT polymorph 1a (Fig. 3b)....."

Fig. R1-1. Structural comparison of the gS87, pS87 and different WT fibril polymorphs. (This figure represents Supplementary Fig. 7 in the revised manuscript.)

a 2D classification averages of straight polymorph (left) and twisted polymorph (right) of pS87 fibril. **b** 2D classification averages of WT polymorph 1a (WT_{1a}). **c**, **d** The structural model of gS87 fibril (**c**) and pS87 fibril (**d**). **e** The structure of unmodified WT α -syn fibrils

(Polymorph 1a, 1b, 2a and 2b) with their PDB codes (top), and the conformation of S87 shown in the zoom-in views (bottom).

2. *Methods for the toxicity assay lack some key details. In the figure legend of Figure 5, it appears that WT, pS87 and gS87 fibrils are at 100 μ M concentration. Is this the monomer equivalent concentration? If so, are there any purification steps performed after fibril preparation? Are the amounts of fibrils normalized for different fibrils? The authors have shown that gS87 and pS87 have lower aggregation propensity than WT, so gS87 and pS87 at the same monomer concentration will have lower amounts of fibrils than WT. Then the difference in toxicity could be caused by the amount of fibrils rather than the structure of the fibrils. Furthermore, there are different fibril polymorphs for WT, gS87, and pS87 fibrils. Have the authors attempted to purify the specific fibril polymorph? If not, how does the presence of different fibril polymorph factor in the interpretation of results?*

REPLY: We extend our gratitude to this reviewer for his/her carefulness and helpful comments. Actually, we have taken α -syn aggregation propensity into consideration before performing the toxicity assay. As the reviewer mentioned, the fibril concentration for the toxicity assay in Fig. 5 refers to the monomer equivalent concentration of α -syn pre-formed fibrils (PFFs). To measure the fibril concentration, the fibril-containing pellet is separated from the monomer-containing supernatant by centrifugation. Then, the fibril-containing pellet was resuspended and its concentration was determined as the fibril concentration. And the amount of WT, gS87 and pS87 fibril were normalized for the following toxicity assay. We added the detailed description in the Method, section “*Preparation of the unmodified WT, gS87 and pS87 α -syn fibrils and PFFs*” on page 14, as following:

“.....For incubated fibrils of unmodified WT, gS87 and pS87 α -syn, the fibril-containing pellet is separated from the monomer-containing supernatant by centrifugation. Then, the fibril-containing pellet was resuspended and its concentration was determined as the fibril concentration.....”

Due to the lack of effective fibril separation method, it is currently impossible to purify certain types of fibril polymorph from the mixture of different fibril polymorphs to investigate their individual function. Consequently, different fibril polymorphs in the mixture contribute to the toxicity results as a whole. Although current methods cannot purify the specific fibril polymorph, our findings suggested that both gS87 and pS87 exhibited an overall reduced neurotoxicity and propagation activity when compared to WT fibrils.

3. *The cryo-EM structures show ordered and disordered regions. Can the authors determine the degree of disorder for the unresolved segments? It would be helpful to determine whether the disordered regions are just barely disordered for cryo-EM or completely disordered. In future studies, it would be helpful to characterize the disordered*

regions with NMR or EPR to get a sense of disorder. In the gS87 structure, the authors identified an ordered island, but stopped short of revealing the identity of these residues. Could these residues be connected to residue 89 via a disordered region?

REPLY: We thank the reviewer for his/her valuable suggestions. Indeed, different disordered regions exhibit different degrees of disorder, ranging from barely disordered to completely disordered. Cryo-EM enables the visualization of the density of the ordered regions. For the disordered regions, no density could be detected because the densities are averaged and could not be distinguished from the solvent. Thus, we are unable to determine the degree of disorder for the unresolved segments based on current cryo-EM data. As suggested by this reviewer, NMR experiment could be used to probe the disorder and structural changes in disordered regions, which will be explored in our future follow-up studies.

In the gS87 structure, the distance between A89 and island is ~ 41 Å, and the distance between E46 and K58 is ~ 40 Å, including 12 residues (Fig. R1-2). Considering the distance, these residues of ordered island could be connected to residue A89 via a disordered region. Unfortunately, it precluded the identification of these residues for the lack of distinct side chain densities¹.

In the revised manuscript, we add Fig. R1-2 as new Supplementary Fig. 6b, and original b, c, d panel in Supplementary Fig. 6 are replaced by d, e, f panel in revised manuscript, respectively. We describe it in Results, section “*Structural analysis of gS87 and pS87 α -syn fibrils*” on page 7, as following:

“.....An unassigned island is observed on the outer surface of the fibril core, adjacent to $\beta 2$ (residues 48-57, Supplementary Fig. 6a). This island is located approximately 41 Å from the C-terminus of residue A89, which is hypothesized to represent a segment of the C-terminal of α -syn (Supplementary Fig. 6b)......”

Fig. R1-2. The unassigned island in gS87 fibril. (This figure represents Supplementary Fig. 6b in the revised manuscript.)

a Cross-section view for the density map with a built-in structure model of gS87 α -syn, and the measured distances between A89 & I50 and E46 & K58 (~40 Å for 12 residues).

4. All the fibrils were prepared *in vitro*. It is important to note the limitations of *in vitro* fibrils as many studies have shown that fibrils isolated/purified from brain tissues adopt different structures from those prepared in the test tubes.

REPLY: We thank this reviewer for raising this important point. We totally agree that there are limitations for fibrils prepared *in vitro* when compared with *ex vivo* fibrils isolated/purified from brain tissues. However, the fibrils deposited in brain are heterogeneous and carry a variety of different PTMs. To pinpoint the effect of each PTM (phosphorylation and O-GlcNAcylation in our case) in regulating protein fibrillation, our semi-synthesis approach which could obtain site-specific PTM modified α -syn provides an effective mean. Nevertheless, following the reviewer's suggestion, we added structural comparison of the fibrils extracted from brains and the fibril reported in our study (Fig. R1-3), and added a sentence in the Discussion section to point out the limitation of our study on page 11, as following:

“.....Both phosphorylation and O-GlcNAcylation at S87 generate fibril structures distinct from previously reported structures of *in vitro* fibril^{44, 50, 55} (Supplementary Fig. 7) and *ex vivo* fibril^{41, 48, 49} (Supplementary Fig. 8). The different conformations of gS87 and pS87 in comparison to *ex vivo* fibrils indicate that these modified forms alone do not replicate the conformations of fibrils extracted from patient brains. It is important to note that the fibrils found in the brain are heterogeneous, encompassing a variety of PTMs. Current *ex vivo* structures may represent the end stage species, and potentially overlook certain minor species. Both O-GlcNAcylation and phosphorylation at S87 mitigate α -syn aggregation, potentially leading to the formation of fibrillar species that do not represent the terminal state in disease contexts. Future investigations focusing on mixed PTMs and the identification of intermediate species during disease progression could yield insights into the impact of PTM cross-talk on α -syn aggregation in real disease conditions.....”

Fig. R1-3. Structural analysis of the gS87, pS87 and *ex vivo* α -syn fibrils. (This figure represents Supplementary Fig. 8 in the revised manuscript.)

a, b The structural model of gS87 fibril (**a**) and pS87 fibril (**b**). **c** The structural model of *ex vivo* fibrils: Lewy fold (PDB: 8A9L), MSA fold (PDB: 6XYO) and JOS fold (PDB: 8BQV). **d** Zoom-in views of conformation of S87 in (**c**).

Some minor points:

1. Figure 1b. TEM picture of seeded gS87 fibrils shows significantly larger fibril width than the wild-type. The scaling should be checked to make sure the gS87 and WT fibrils are shown at the same magnification.

REPLY: Thanks to the reviewer for his/her carefulness. Following the reviewer's suggestion, the scaling has been repeatedly and carefully checked to make sure the magnification is same. Additionally, the NS-TEM image of unmodified WT fibrils (adding 1 mol% PFF) was updated with better negative-staining (Fig. R1-4). Meanwhile, the fibril width of gS87 fibril (~12 nm) (Fig. 2a) is larger than WT fibril⁴ (~10 nm), resulting in the seeded gS87 fibrils displaying a larger fibril width than WT.

Fig. R1-4. Fibril characterization of gS87, pS87 and unmodified WT α -syn. (This figure represents new Fig. 1b in the revised manuscript.)

a Left: ThT kinetic assay (top) and NS-TEM images (bottom) of unmodified WT, gS87, and pS87 α -syn fibrils. Right: ThT kinetic assay (top) and NS-TEM images (bottom) of unmodified WT, gS87 and pS87 α -syn fibrils in the presence of 1 mol% PFF formed by the unmodified WT α -syn monomer. Zoom-in views of gS87 and pS87 ThT kinetic assay were shown. The fibrils were characterized by NS-TEM at the endpoint (60h) of the ThT kinetic assay. Data correspond to mean \pm s.d., n=3. Scale bar: 200 nm.

2. Figure 3. Suggest to flip the WT fibril structure (Figure 3b) vertically, so that the orientation resembles gS87. In the flipped orientation, it is more obvious that the WT fibril and gS87 have an overall resemblance at the backbone level even though the detailed packing is different.

REPLY: We thank this reviewer's insightful suggestions on Fig. 3b. The WT fibril structure has been flipped vertically to highlight the overall resemblance at the backbone level (Fig. R1-5).

Fig. R1-5. The flipped structure of unmodified WT α -syn fibril. (This figure represents new Fig. 3b in the revised manuscript.)

a The structure of unmodified WT α -syn fibril, with the conformation of S87 and the salt bridge between K80 and E46 shown in the zoom-in views.

3. *ThT* was never spelled out.

REPLY: Thanks for the reviewer’s correction. We have corrected “ThT” into “thioflavin T (ThT)” in the Results, section “*Characterization of the gS87 and pS87 α -syn fibrils*” on page 5 in revised manuscript, as following:

“We next investigated the influence of glycosylation and phosphorylation at S87 on α -syn fibrillation through thioflavin T (ThT) kinetic assay and negative-staining (NS) transmission electron microscopy (TEM).....”

Reviewer #2 (Remarks to the Author):

This study looks at α -syn fibrils with phosphorylation and O-GlcNAcylation at the site S87. This produces distinct different fibril morphologies than the unmodified protein and also reduces neurotoxicity and propagation of fibril compared to unmodified protein. This helps the further understanding of posttranslational modifications in amyloid diseases.

Overall, I support the publication of the manuscript. The methodology is sound and meets the expected standards.

REPLY: We express our gratitude to the reviewers for their insightful remarks and acknowledgment of the significance of our research. We carefully addressed the reviewer's concerns below and revised the manuscript accordingly.

But I have some comments and questions for the authors:

How would the modification at position S87 impact the structures of ex vivo α -syn fibrils (Lewy fold, MSA folds)? A discussion or comparison to the ex vivo fibril structures in relation to your findings would be appreciated.

REPLY: We appreciate the reviewer very much for this important suggestion. Taken the reviewer's suggestion, a discussion and structural comparison to the *ex vivo* fibril structures have been added to Discussion of the revised manuscript. Interestingly, the *ex vivo* fibrils from brain tissues and gS87, pS87 fibrils showed different structures at backbone level. S87 in Lewy fold α -syn fibril¹ (PDB: 8A9L) forms a steric-zipper interaction with other residues. However, in MSA fold⁷ (PDB: 6XYO) and JOS fold⁸ (PDB: 8BQV), S87 does not form any direct interactions with other residues.

In the revised manuscript, we add Fig. R2-1 as Supplementary Fig. 8, and described it in Results section "*Comparison of α -syn fibril structures with two different modifications*" on page 8 in revised manuscript, as following:

".....In the WT polymorph 1a, MSA fold and Juvenile-onset synucleinopathy (JOS) fold^{44, 48, 49}, residue S87 is not involved in direct interactions with other residues (Fig. 3b, Supplementary Fig. 8c, d). Conversely, in WT polymorphs 2a, 2b and Lewy fold^{41, 50}, S87 engages in zipper-like interactions (Supplementary Fig. 7e, 8c, d)....."

As well as in Discussion section on page 10, as following:

“.....Both phosphorylation and O-GlcNAcylation at S87 generate fibril structures distinct from previously reported structures of *in vitro* fibril^{44, 50, 55} (Supplementary Fig. 7) and *ex vivo* fibril^{41, 48, 49} (Supplementary Fig. 8). The different conformations of gS87 and pS87 in comparison to *ex vivo* fibrils indicate that these modified forms alone do not replicate the conformations of fibrils extracted from patient brains. It is important to note that the fibrils found in the brain are heterogeneous, encompassing a variety of PTMs. Current *ex vivo* structures may represent the end stage species, and potentially overlook certain minor species. Both O-GlcNAcylation and phosphorylation at S87 mitigate α -syn aggregation, potentially leading to the formation of fibrillar species that do not represent the terminal state in disease contexts. Future investigations focusing on mixed PTMs and the identification of intermediate species during disease progression could yield insights into the impact of PTM cross-talk on α -syn aggregation in real disease conditions.....”

Fig. R2-1. Structural analysis of the gS87, pS87 and *ex vivo* α -syn fibrils. (This figure represents Supplementary Fig. 8 in the revised manuscript.)

a, b The structural model of gS87 fibril (**a**) and pS87 fibril (**b**). **c** The structural model of *ex vivo* fibrils: Lewy fold (PDB: 8A9L), MSA fold (PDB: 6XYO) and JOS fold (PDB: 8BQV). **d** Zoom-in views of conformation of S87 in (**c**).

Minor comments and questions:

A proteolysis experiment of unmodified and modified α -syn fibrils could be interesting, because stability against proteases seems to be a major factor for which morphologies are found in vivo.

REPLY: We greatly appreciate this valuable suggestion. Taken the reviewer's suggestion, the proteinase K (PK) digestion experiment of unmodified and modified α -syn fibrils was done to verify their stabilities against proteases. The results showed that pS87 PFFs had the worst stability, when gS87 PFFs were digested slightly faster than WT PFFs (Fig. R2-2). This result has been added to Results, section "*Characterization of the gS87 and pS87 α -syn fibrils*" on page 5 in revised manuscript, as following:

".....Additionally, we assessed the stability of these fibrils under proteinase K (PK) digestion. The results demonstrated that both gS87 and pS87 variants were digested more rapidly compared to the unmodified WT PFFs. Notably, pS87 fibrils exhibited the lowest stability (Supplementary Fig. 9b)....."

Fig. R2-2. PK digestion of WT, gS87 and pS87 PFFs. (This figure represents Supplementary Fig. 9b in the revised manuscript.)

a WT, gS87 and pS87 PFFs were incubated with proteinase K with different concentrations as indicated at 37°C for 20 min.

Did you do polishing in Relion after 3D auto-refinement?

REPLY: We didn't do polishing in Relion after 3D auto-refinement.

One or two decimal points for the B-factor in Supplementary Table 1 are enough.

REPLY: Thanks for the reviewer's suggestion. As suggested, we retained two decimal points for "Map sharpening B-factor (\AA^2)" in Supplementary Table 1.

Clash score is pretty high for the estimated resolution. It should be possible to get below 5.

REPLY: We thank this reviewer for raising this very helpful suggestion. Taken the reviewer's suggestion, the models of gS87 and pS87 have been modified several rounds to make their clash scores below 5. Moreover, all description related to gS87 and pS87 structures in manuscript and supplementary information, including new Fig. 2 (Fig. R2-3), new Fig. 3 (Fig. R2-4), new Fig. 4 (Fig. R2-5), new Supplementary Fig. 6, (Fig. R2-6), and the "Atomic model" section in new Supplementary Table 1 (Table R2-1), are re-edited based on the newly modified models.

Fig. R2-3. Cryo-EM structures of gS87 and pS87 fibrils. (This figure represents new Fig. 2 in the revised manuscript.)

a, b The density maps of the gS87 (**a**) and pS87 (**b**) fibril are colored in light-blue and pink, respectively. Fibril parameters including half-pitch, fibril width, twist angle, and helical rise are marked. **c, d** Cross-section view for the density maps with a built-in structure model of gS87 (**c**) and pS87 (**d**) α -syn. Topology diagrams are shown at the bottom right. **e** Zoom-in views of the GlcNAc molecules in the electron density from (**c**) are shown. **f, g** Views of three layers of gS87 (**f**) and pS87 (**g**) α -syn fibrils are shown in the cartoon. The β -strands of the fibril structures are numbered and labeled accordingly with the fibril axis indicated.

Fig. R2-4. Structural analysis of gS87, pS87, unmodified WT, and pY39 α -syn fibril. (This figure represents new Fig. 3 in the revised manuscript.)

a The structural model of gS87 fibril, with the zoom-in views the interactions between GlcNAc, K80, E61, T81, V82, I88 and A89, and the hydrophobic zipper-like interactions with the involved residues labeled. **b** The structure of unmodified WT α -syn fibril, with the conformation of S87 and the salt bridge between K80 and E46 shown in the zoom-in views. **c** The structure of pS87 fibril, with the hydrophobic interactions of the interface between two protofilaments, and two pairs of salt bridges shown in the zoom-in views. Residues involved in the inter-protofilamental interactions are shown in spheres. **d** The structure of pY39 fibril with the electrostatic interactions of the phosphate group bound to Y39 and K21, K32 and K34 shown in the zoom-in view. Distances are shown in Å. The PDB code of each fibril structure is indicated.

Fig. R2-5. Comparison of α -syn fibril structures with different modifications. (This figure represents new Fig. 4 in the revised manuscript.)

a one-layer models of unmodified WT (PDB: 6A6B), gS87 (PDB:8JEX), pS87 (PDB:8JEY) and pY39 (PDB: 6L1T) fibrils with the N-terminal region colored in blue, the NAC colored in yellow and the C-terminal region colored in red. **b** The secondary structure alignment of four α -syn fibril structures from (a) with different colors for three regions. **c** Schematic diagram shows that both same PTM modified at different sites and different PTMs modified at same site induce distinct fibril core structures.

Fig. R2-6. Structural analysis of the gS87 and pS87 α -syn fibrils. (This figure represents new Supplementary Fig. 6 in the revised manuscript.)

a In the electron density map of the gS87 fibril, the unassigned island was observed on the outer surface of the fibril core, which was adjacent to β 2. **b** Cross-section view for the density map with a built-in structure model of gS87 α -syn, and the measured distances between A89 & island and E46 & K58 (~ 40 Å for 12 residues). **c** gS87 density maps of different threshold values with extra densities marked. **d** Zoom-in views of hydrophilic zipper-like interactions in gS87 fibril structure to the stabilization of the U-shaped structure. **e** Zoom-in views of the hydrophobic interactions in gS87 fibril model. Residues involved in the interactions are indicated in spheres. **f** The structure of the pS87 fibril, with the salt bridge between K32 and E46, the hydrogen bond between K34 and Y39, and the steric zipper-like hydrophobic interaction shown in the zoom-in views.

Table R2-1. Cryo-EM data collection, modeling and refinement statistics. (This figure represents new Supplementary Table 1 in the revised manuscript.)

Data collection and processing	gS87 α-syn (EMD: 36202) (PDB: 8JEX)	pS87 α-syn (EMD: 36203) (PDB: 8JEY)
Data Collection		
Magnification (\times)	105,000	105,000
Pixel size (Å)	0.83	0.83
Defocus Range (μ m)	-1.0 to -2.0	-1.0 to -2.0
Voltage (kV)	300	300
Camera	BioContinuum K3	BioContinuum K3
Microscope	Krios G4	Krios G4
Exposure time (s/frame)	0.05	0.05
Number of frames	40	40
Total dose ($e^-/\text{Å}^2$)	55	55
Reconstruction		
Micrographs	2,134	2,423
Manually picked fibrils	21,328	27,806
Box size (pixel)	360	360
Inter-box distance (Å)	30	30
Initial particle images (no.)	465,930	647,678
Final particle images (no.)	24,910	61,047
Resolution (Å)	3.1	2.6
Map sharpening B-factor (Å ²)	-96.50	-86.47
Helical rise (Å)	-179.72	-179.72
Helical twist (°)	2.41	2.41
Atomic model		
Non-hydrogen atoms	2,616	3,060
Protein residues	384	432
Ligands	6	0
r.m.s.d. Bond lengths	0.008	0.004
r.m.s.d. Bond angles	0.986	0.746
All-atom clash score	4.96	4.75
Rotamer outliers	0 %	0 %
Ramachandran Outliers	0 %	0 %

Ramachandran Allowed	3.33 %	4.29 %
Ramachandran Favored	96.67 %	95.71 %

Reviewer #3 (Remarks to the Author):

The manuscript by Hu et al describes structural studies of two post-translationally modified alpha synuclein fibrils at a single site, S87, and their pathologically relevant effects on the propagation of amyloid aggregation and neurotoxicity in primary neurons. Chemical ligation and synthesis yielded high purity alpha synuclein proteins with single site modification, which allowed studies of site-specific effects. Using cryo-TEM 3D reconstruction, both fibril structures were obtained at ~3 angstrom resolution, where distinct features between these PTM variants, and between PTM and wt alpha synuclein fibrils were identified. Potentially, the work may prompt the current understanding of the influence of site-specific PTM of amyloid proteins on their pathological roles. In the current version, some of the experiments/interpretation of results may be improved:

REPLY: We thank the reviewer very much for recognizing the importance and significance of our work. And we highly value his/her insightful comments and suggestions provided. We carefully addressed the reviewer's comments below and revised the manuscript accordingly.

1. Biophysical characterization of fibrils.

(1) ThT-fluorescence: Please provide the kinetics traces for gS87 and pS87 fibrillation that show clear increment of fluorescence emission, especially for pS87.

REPLY: We thank this reviewer for this important question about ThT kinetic assay. Following the reviewer's suggestion, the kinetics traces for gS87 and pS87 fibrillation that show clear increment of fluorescence emission have been added into the Fig. 1b (Fig. R3-1). As the results shown, both gS87 and pS87 α -syn exhibited significantly reduced capability for fibrillation compared to unmodified WT α -syn. In the revised manuscript, we add Fig. R3-1 as new Fig. 1b with two zoom-in views of ThT kinetic assay.

Fig. R3-1. Fibril characterization of gS87, pS87 and unmodified WT α -syn. (This figure replaces original Fig. 1b in the revised manuscript.)

a Left: ThT kinetic assay (top) and NS-TEM images (bottom) of unmodified WT, gS87, and pS87 α -syn fibrils. Right: ThT kinetic assay (top) and NS-TEM images (bottom) of unmodified WT, gS87 and pS87 α -syn fibrils in the presence of 1 mol% PFF formed by the unmodified WT α -syn monomer. Zoom-in views of gS87 and pS87 ThT kinetic assay were shown. The fibrils were characterized by NS-TEM at the endpoint (60h) of the ThT kinetic assay. Data correspond to mean \pm s.d., n=3. Scale bar: 200 nm.

(2) Negatively stained TEM: Please provide TEM images that shows fibrils morphologies for gS87 and pS87, especially for pS87. It is unclear from the current images whether or not there are fibrils. It is expected from the text that both fibrils should have twisted morphologies on their filaments.

REPLY: Thanks for the comments. Following the reviewer's suggestion, the NS-TEM and cryo-EM images were added as the Supplementary Fig. 9a (Fig. R3-2), revealing the twisted morphologies in both gS87 and pS87 fibrils, especially evident in the characterization of cryo-EM.

In the revised manuscript, we add Fig. R3-2 as Supplementary Fig. 9a in Results, section "*Characterization of the gS87 and pS87 α -syn fibrils*" on page 5, as following:

".....In sharp contrast, gS87 α -syn started to form fibril after 20 hours incubation. The ThT signal of pS87 sample slowly picked up after 40 hours with much less fibril formed as revealed by NS-TEM (Fig. 1b, Supplementary Fig. 9a)....."

Fig. R3-2. NS-TEM and cryo-EM characterization of the gS87 and pS87 fibrils. (This figure represents Supplementary Fig. 9a in the revised manuscript.)

a The fibrils of gS87 (left panel) and pS87 (right panel) fibrils characterized by NS-TEM (top) and cryo-EM (bottom) with twisted fibrils marked. Scale bar: 100 nm.

(3) *AFM: For gS87 and pS87, it seems that the pitches are heterogenous, are there any quantitative or semi-quantitative analysis of the pitch lengths based on the AFM? And how would these analyses related to the cryo-TEM structures?*

REPLY: We appreciate very much of the thoughtful suggestion from this reviewer. Taken the reviewer’s suggestion, we did statistical analysis of the AFM results of the gS87 and pS87 fibrils (Fig. R3-3), and the average half-pitch lengths are ~156 nm and ~157 nm respectively, with subtle differences in specific fibril. In the cryo-EM datasets, the half-pitch lengths are ~156 nm and ~154 nm respectively, based on 21,328 fibrils for gS87 and 27,806 fibrils for pS87. Therefore, the AFM analysis are similar to the results of cryo-EM dataset, which can be used as initial parameters during 3D reconstruction.

In the revised manuscript, we add Fig. R3-3 as Supplementary Fig. 9c, d, and described it in Results, section “*Characterization of the gS87 and pS87 α -syn fibrils*” on page 5, as following:

“.....*Statistical analysis of AFM data revealed that the average half-pitch lengths of gS87 α -syn and pS87 α -syn fibrils are approximately 156 nm and 157 nm (Supplementary Fig. 9c, d).*”

Fig. R3-3. AFM characterization and statistics of gS87 and pS87 fibrils. (This figure represents Supplementary Fig. 9c, d in the revised manuscript.)

a The AFM statistics of gS87 and pS87 half-pitch length with the mean of 156 nm, 157 nm, respectively. Data correspond to mean \pm s.d., $n = 31$ (gS87), $n = 40$ (pS87). **b** AFM characterization gS87 and pS87 fibrils with half-pitch marked. Scale bar: 200 nm.

2. Cryo-TEM structures.

(1) There is a large gap in its quaternary interface for gS87 (the authors have also mentioned this in line 162-166, about 15 angstrom), without any identified electron density. What may be the interactions to hold this interface?

REPLY: We appreciate this reviewer for raising this great point. The distance of the large gap exceeds $\sim 15 \text{ \AA}$ (Fig. 2a), suggesting a relatively weak interaction between the two protofilaments. As for the interaction holding the two protofilament together, there could be two possibilities. (1) considering that the fibril was twisted rather than straight, two protofilaments are entangled together, which may contribute to maintaining the interface. (2) gS87 density maps with different thresholds showed that there are two extra but very weak densities in its interface (Fig. R3-4), which may represent some molecules such as phosphate group that bridge the interface together. Due to the presence of charged residues (Y39, K43, K45 and E46) on the interface and phosphate group contained in the buffer, electrostatic interactions mediated by phosphate groups may be involved in maintaining the stability of the interface.

In the revised manuscript, we add Fig. R3-4 as Supplementary Fig. 6c, and described it in Results, section “Structural analysis of gS87 and pS87 α -syn fibrils” on page 7, as following:

“.....Two additional, albeit weak, densities were identified at the protofilamental interface (Supplementary Fig. 6c), which are hypothesized to represent solvent molecules that potentially bridge the interface together.”

Fig. R3-4. gS87 density maps of different threshold values with extra densities marked. (This figure represents Supplementary Fig. 6c in the revised manuscript.)

(2) *It is mentioned that for pS87 fibrils, 74% of the fibrils have straight polymorph. Why was the 3D reconstruction done for the minor 26% twisted polymorph?*

REPLY: Thanks for the reviewer’s comment. Unfortunately, due to the technical limitations of cryo-EM helical reconstruction, it’s impossible to determine the structure of straight fibrils so far^{1, 2, 3}. As a result, only the structure of the twisted filament polymorph (~26%) was determined in pS87 fibrils. We also look forward to developing new methods that will allow us to determine the structure of straight fibrils in the future.

(3) *It is stated in line 197-199: GlcNAc at S87 introduces new interactions with K80 and E61, leading to a structural rearrangement.... What are the evidence that the structural difference between gS87 and wt fibrils are actually driven by this site-specific PTM? In other words, do the interactions between GlcNAc-S87 and the surrounding residues occur prior to the formation of the rest of fibrillar core?*

REPLY: We thank this reviewer for the insightful question. We agree with the reviewer that we currently don’t have direct experimental evidence supporting that the structural rearrangement is driven by this site-specific PTM, since cryo-EM can only give us the final structure of the fibril but not the folding intermediate in this study. Thus, we don’t know whether the S87-associated structural motif is formed prior to the formation of the rest structure.

Accordingly, We have revised the related description from “*Therefore, the GlcNAc modified at S87 introduces new interactions with K80 and E61, leading to structural rearrangement of α -syn and the formation of a distinct β -strand pattern to create a new fibril core structure (Fig. 4a, b).*” into “*Therefore, the GlcNAc modified at S87 formed new interactions with K80 and E61, accompanied by the structural rearrangement of α -syn and the formation of a distinct β -strand pattern to create a new fibril core structure (Fig. 4a, b).*” on page 8 in revised manuscript.

In addition, we added a description in the Discussion section on page12, as following:

“.....Of note, cryo-EM structure captures α -syn fibrils in their final aggregated state. Consequently, it remains unclear whether PTM-mediated interactions occur prior to or concurrent with the formation of the rest of the fibril cores. To elucidate this, further studies are required to characterize the kinetic intermediates of protein fibrillar assembly at the atomic level.....”

(4) The discussion of pS87 effect on fibril structures in line 224-232 is not convincing because Fig. 3b shows that in wt alpha synuclein fibril, S87 is not involved in any tertiary or quaternary interactions. Why does this site-specific PTM have such a significant impact on the structure?

REPLY: We thank the reviewer for pointing this out. In the structure of WT α -syn, S87 does not form any direct interactions with other residues (Fig. 3b). After phosphorylation at S87, there are not enough positively charged residues to stabilize the phosphate group in pS87 structure, which is different from phosphorylation at Y39. As a consequence, the C-terminal region of NAC remains flexible and don't participate in the fibril core formation. The PTM on specific site may influence the initial structural folding⁹, including stabilizing certain types of interaction (PTM engages into fibril core, such as gS87 α -syn and pY39 α -syn), and avoiding other interactions (PTM may be located in the disordered region, such as pS87 α -syn).

3. Immunostaining.

What is the justification for using the pS129-tau-specific antibody, but not total tau antibody, or both?

REPLY: Thanks for the reviewer's question. Ser129 is extensively phosphorylated in Parkinson's disease (PD), dementia with Lewy bodies (DLB) and multiple system atrophy (MSA)^{10, 11}, as well as in transgenic animal models of synucleinopathies^{12, 13}. Therefore, Ser129 phosphorylation (pS129) is a hallmark of pathological α -syn^{10, 14, 15} and its levels increased after treatment of α -syn PFF^{16, 17, 18}. In contrast, the relevance between total α -syn and α -syn pathology is not well established. Therefore, pS129 α -syn antibody was used to test the propagation activity, as commonly used in other related papers^{4, 15, 19, 20, 21, 22, 23}.

4. Minor points

(1) Line 50: Please give the full name of MSA.

REPLY: We thank this reviewer for the correction. The spelling of “MSA” has been replaced with “multiple system atrophy (MSA)” in the manuscript on page 3.

(2) Fig. 2b: Should the helical rise be 2.41 angstrom instead of 4.82 for pS87?

REPLY: We appreciate this reviewer for his/her carefully reading. As shown in Fig. R3-5, a helical rise is 2.41 Å and double helical rises is 4.82 Å after the application of P2₁ symmetry in 3D reconstruction. In fact, the helical rise of pS87 shown in the Fig. 2b is actually double helical rises for a better display.

Fig. R3-5. gS87 and pS87 density maps with a helical rise (left) and double helical rises (right).

References

1. Yang Y, *et al.* Structures of α -synuclein filaments from human brains with Lewy pathology. *Nature* **610**, 791-795 (2022).
2. Shi Y, *et al.* Structure-based classification of tauopathies. *Nature* **598**, 359-363 (2021).
3. Li D, Liu C. Structural Diversity of Amyloid Fibrils and Advances in Their Structure Determination. *Biochemistry* **59**, 639-646 (2020).
4. Li Y, *et al.* Amyloid fibril structure of alpha-synuclein determined by cryo-electron microscopy. *Cell Res* **28**, 897-903 (2018).
5. Li B, *et al.* Cryo-EM of full-length α -synuclein reveals fibril polymorphs with a common structural kernel. *Nature Communications* **9**, 3609 (2018).
6. Guerrero-Ferreira R, *et al.* Two new polymorphic structures of human full-length alpha-synuclein fibrils solved by cryo-electron microscopy. *Elife* **8**, e48907 (2019).
7. Schweighauser M, *et al.* Structures of α -synuclein filaments from multiple system atrophy. *Nature* **585**, 464-469 (2020).

8. Yang Y, *et al.* New SNCA mutation and structures of α -synuclein filaments from juvenile-onset synucleinopathy. *Acta Neuropathologica* **145**, 561-572 (2023).
9. Li D, Liu C. Hierarchical chemical determination of amyloid polymorphs in neurodegenerative disease. *Nature Chemical Biology* **17**, 237-245 (2021).
10. Fujiwara H, *et al.* alpha-Synuclein is phosphorylated in synucleinopathy lesions. *Nat Cell Biol* **4**, 160-164 (2002).
11. Colom-Cadena M, *et al.* Synaptic phosphorylated α -synuclein in dementia with Lewy bodies. *Brain* **140**, 3204-3214 (2017).
12. Oueslati A. Implication of Alpha-Synuclein Phosphorylation at S129 in Synucleinopathies: What Have We Learned in the Last Decade? *Journal of Parkinson's Disease* **6**, 39-51 (2016).
13. Chen L, Feany MB. α -Synuclein phosphorylation controls neurotoxicity and inclusion formation in a Drosophila model of Parkinson disease. *Nature Neuroscience* **8**, 657-663 (2005).
14. Gorbatyuk OS, *et al.* The phosphorylation state of Ser-129 in human α -synuclein determines neurodegeneration in a rat model of Parkinson disease. *Proceedings of the National Academy of Sciences* **105**, 763-768 (2008).
15. Peng C, *et al.* Cellular milieu imparts distinct pathological alpha-synuclein strains in alpha-synucleinopathies. *Nature* **557**, 558-563 (2018).
16. Mao X, *et al.* Pathological alpha-synuclein transmission initiated by binding lymphocyte-activation gene 3. *Science* **353**, aah3374 (2016).
17. Volpicelli-Daley Laura A, *et al.* Exogenous α -Synuclein Fibrils Induce Lewy Body Pathology Leading to Synaptic Dysfunction and Neuron Death. *Neuron* **72**, 57-71 (2011).
18. Luk KC, *et al.* Pathological α -Synuclein Transmission Initiates Parkinson-like Neurodegeneration in Nontransgenic Mice. *Science* **338**, 949-953 (2012).
19. Zhang S, *et al.* Post-translational modifications of soluble alpha-synuclein regulate the amplification of pathological alpha-synuclein. *Nat Neurosci* **26**, 213-225 (2023).
20. Zhao K, *et al.* Parkinson's disease-related phosphorylation at Tyr39 rearranges alpha-synuclein amyloid fibril structure revealed by cryo-EM. *Proc Natl Acad Sci U S A* **117**, 20305-20315 (2020).
21. Kim S, *et al.* Transneuronal Propagation of Pathologic α -Synuclein from the Gut to the Brain Models Parkinson's Disease. *Neuron* **103**, 627-641.e627 (2019).
22. Butler YR, *et al.* α -Synuclein fibril-specific nanobody reduces prion-like α -synuclein spreading in mice. *Nature Communications* **13**, 4060 (2022).
23. Kam T-I, *et al.* Amelioration of pathologic α -synuclein-induced Parkinson's disease by irisin. *Proceedings of the National Academy of Sciences* **119**, e2204835119 (2022).

Reviewer #1 (Remarks to the Author):

The authors have done an excellent job to fully address my concerns. I recommend this manuscript to be accepted for publication.

Reviewer #2 (Remarks to the Author):

I have no further comments and support the publication of the manuscript.

Reviewer #3 (Remarks to the Author):

I believe the authors have adequately addressed my questions and suggestions. The inclusion of additional biophysical data and analyses is appreciated. The authors have also provided reasonable explanations and revisions regarding the questions related to cryo-EM structures, which can be not fully addressed due to the current limitations. I recommend publishing the manuscript in Nature Communications.